# Participation bias in the UK Biobank distorts genetic associations and downstream analyses

**Tabea Schoeler** [1,2] ✉, **Doug Speed** [3], **Eleonora Porcu**[4], **Nicola Pirastu**[5], **Jean-Baptiste Pingault**[2,6] **& Zoltán Kutalik** [1,7,8] ✉

While volunteer-based studies such as the UK Biobank have become the cornerstone of genetic epidemiology, the participating individuals are rarely representative of their target population. To evaluate the impact of selective participation, here we derived UK Biobank participation probabilities on the basis of 14 variables harmonized across the UK Biobank and a representative sample. We then conducted weighted genome-wide association analyses on 19 traits. Comparing the output from weighted genome-wide association analyses ($n_{effective}$ = 94,643 to 102,215) with that from standard genome-wide association analyses ($n$ = 263,464 to 283,749), we found that increasing representativeness led to changes in SNP effect sizes and identified novel SNP associations for 12 traits. While heritability estimates were less impacted by weighting (maximum change in $h^2$, 5%), we found substantial discrepancies for genetic correlations (maximum change in $r_g$, 0.31) and Mendelian randomization estimates (maximum change in $\beta_{STD}$, 0.15) for socio-behavioural traits. We urge the field to increase representativeness in biobank samples, especially when studying genetic correlates of behaviour, lifestyles and social outcomes.

The overarching aim of genetic epidemiology is to elucidate the genetic underpinning of health and disease. To maximize power for genome-wide discovery, researchers curate large biobanks with rich genetic and phenotypic data. To ensure the validity of findings in genome-wide association (GWA) studies, researchers aim to eliminate potential sources of bias, such as population stratification, assortative mating, measurement error and indirect genetic effects[1-4].

A particularly challenging bias that is typically not considered in genetic studies can occur when biobanks collect data from individuals that are not representative of their target population[5-7]. Under certain conditions, research on non-representative samples can lead to valid conclusions—for example, when study participation is unrelated to both the independent and dependent variables. However, many commonly studied factors influence study participation. These may include mental and physical health, substance use (such as cigarettes and alcohol), income, and educational attainment[8-12], where study participants are typically healthier than the target population. Such 'healthy-volunteer bias' is well documented in the UK Biobank (UKBB), one of the most widely used resources for biomedical research. Of the nine million people invited to take part in the UKBB, only 5.5% (~500,000) participated in the study—a sample of volunteers with healthier lifestyles, higher levels of education and better health than the general UK population[13,14].

[1]Department of Computational Biology, University of Lausanne, Lausanne, Switzerland. [2]Department of Clinical, Educational and Health Psychology, University College London, London, UK. [3]Quantitative Genetics and Genomics, Aarhus University, Aarhus, Denmark. [4]Precision Medicine Unit, Biomedical Data Science Center, Lausanne University Hospital and University of Lausanne, Lausanne, Switzerland. [5]Genomics Research Centre, Human Technopole, Milan, Italy. [6]Social, Genetic and Developmental Psychiatry Centre, Institute of Psychiatry, Psychology and Neuroscience, King's College London, London, UK. [7]Swiss Institute of Bioinformatics, Lausanne, Switzerland. [8]University Center for Primary Care and Public Health, Lausanne, Switzerland. ✉e-mail: tabea.schoeler@unil.ch; zoltan.kutalik@unil.ch

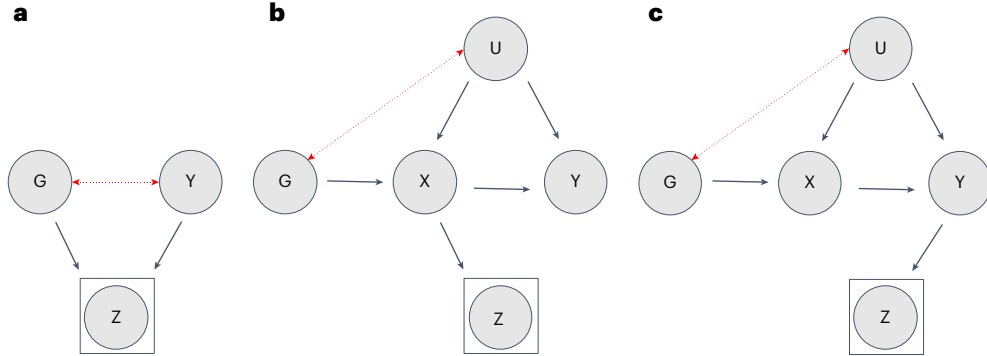

**Fig. 1 | The impact of participation bias in genetic studies. a–c,** The relationships between a genetic variant (G), an exposure (X) or outcome (Y), and study participation (Z). Panel **a** illustrates the effect of participation bias in GWA studies, where Z is a common consequence of G and Y (red dotted line). Conditioning on a common consequence (Z) induces a non-causal association between G and Y. Panels **b,c** illustrate the effect of participation bias in MR studies, where bias occurs if Z is a consequence of either X (**b**) or Y (**c**). Conditioning on Z induces an association between the genetic variant and confounders, thereby violating the MR assumption of exchangeability. This figure is a simplified illustration of how participation bias can impact results obtained from two commonly employed methods in genomic studies. For further examples illustrating the impact of selection bias, see Hernán et al.[7].

Given the growing reliance on non-representative biobanks, it is paramount to assess the extent to which study participation induces bias in genome-wide studies and downstream analyses. In observational studies using UKBB data, participation bias has already been shown to distort phenotypic exposure–outcome associations[12,13,15]. If study participation includes a genetic component, biased estimates are also expected in genetic studies[16]. In gene-discovery studies, non-random participation may distort the association between a genetic variant and the outcome (Fig. 1a). In Mendelian randomization (MR) (a causal inference technique using single nucleotide polymorphisms (SNPs) as instrumental variables), participation bias could induce an association between genetic instruments and unmeasured confounders of the exposure–outcome relationship, thereby violating a key assumption of the method (Fig. 1b,c). Recent genome-wide studies investigating proxies of participation bias have already described genetic variation associated with participation and questionnaire responding[17–24], indicating that genetic studies are not immune to bias. While much of the recent GWA output has been produced by non-representative biobanks (for example, UKBB, Million Veteran Program and 23andMe), the extent to which participation bias affects gene discovery and downstream analyses is currently unknown.

Researchers can correct for participation bias by the use of samples that are representative of their target population—a broader group from which a study sample is drawn and to which the study results should generalize. In case of the UKBB, the target population is middle-aged to older adults of recent European ancestry living in the United Kingdom, which is not the same as the general UK population (Supplementary Information). Here we derive a model for participation probability and create a pseudo-sample of the UKBB matching its target population with respect to 14 variables. We can thereby evaluate how a shift towards representativeness impacts genome-wide findings and downstream analyses. We anticipate that these findings will help characterize the impact of participation bias in large volunteer-based samples used for biomedical research and help pin down areas of research that might be particularly susceptible to bias when relying on non-representative samples.

## Results

### Samples

From the five Health Survey England (HSE) cohorts comprising a total sample of $n = 81,118$, we retained $n = 22,646$ after applying the same inclusion criteria used for UKBB recruitment (Methods). After further exclusion of HSE individuals with missing data on the 14 auxiliary variables, we included a final sample of $n = 21,816$. Comparing the distribution of a subset of auxiliary variables also available in the UK Census Microdata ($n = 895,649$) shows that the profile of the HSE sample closely matches that of the Census sample (Supplementary Table 1). More specifically, proportions were comparable between the HSE and Census but deviated in the UKBB for most of the selected variables, such as proportion ($P$) of female gender ($P_{CENSUS} = 51\%$, $P_{HSE} = 51\%$, $P_{UKBB} = 54\%$), proportion of individuals of age $\geq 65$ ($P_{CENSUS} = 13\%$, $P_{HSE} = 13\%$, $P_{UKBB} = 19\%$), mean ($M$) age when individuals completed full-time education ($M_{CENSUS} = 16.6$, $M_{HSE} = 16.4$, $M_{UKBB} = 17.2$) and proportion of retired individuals ($P_{CENSUS} = 19\%$, $P_{HSE} = 19\%$, $P_{UKBB} = 34\%$). Further inspection of the associations between variables available in the HSE and UK Census (Supplementary Fig. 1) highlights that the HSE captures the characteristics of the population residing in England well.

Of the initial UKBB sample (502,645 participants), we excluded individuals of age >69 and <40 ($n = 2463$), individuals from Scotland or Wales ($n = 56,483$), individuals who self-identify as non-white ($n = 28,371$) and individuals withdrawing consent ($n = 161$). We further removed 21,868 (5.27%) individuals with missing data for any of the auxiliary variables. Since these individuals can be considered a special case of missingness due to non-participation, which the probability weights were designed to compensate for, we did not impute missing data for the auxiliary variables. The sampling weights were generated for $n = 393,299$ UKBB individuals, of which 109,550 were removed after we applied quality control steps for genome-wide analyses (Methods).

### Performance of the UKBB probability weights

We derived a model for participation probability by comparing 14 harmonized characteristics of UKBB participants with those of a representative sample (HSE). The application of the resulting probability weights then facilitates the creation of a (weighted) pseudo-sample of the UKBB that is more representative of its (representative) target population (HSE). Figure 2a shows the distribution of the normalized probability weights ($w_{in}$) for UKBB individuals. We obtained the probabilities used to construct the weights from a LASSO regression model retaining 454 of the 903 initially included predictors. Figure 2b illustrates which auxiliary variables were the most strongly linked to UKBB participation (UKBB = 1; HSE = 0), highlighting that older (retired), more educated and non-smoking people were particularly likely to participate.

To evaluate the performance of the weights, we first assessed whether probability weighting recovered the reference (HSE) population distributions. We included the generated weights in a univariate logistic regression model predicting UKBB participation, where UKBB individuals were given their normalized weight ($w_{in}$) and HSE

participants were given a weight of 1. When we applied probability weighting (shown on the right side of Fig. 2b), previously significant predictors became non-significant. All means and proportions in the HSE, UKBB (unweighted) and UKBB (weighted) are provided in Supplementary Table 2.

Next, we estimated the degree of bias reduction in our 14 variables following probability weighting. Here we quantified participation bias as the difference between an estimate of association obtained in the UKBB ($r_{UKBB}$) and the reference sample ($r_{HSE}$). The largest differences ($r_{diff} = |r_{HSE} - r_{UKBB}|$) were for employment status with overall health ($r_{diff} = 0.19$; $r_{HSE} = -0.25$; $r_{UKBB} = -0.06$), overall health with age ($r_{diff} = 0.12$; $r_{HSE} = -0.13$; $r_{UKBB} = -0.01$), household size with income ($r_{diff} = 0.10$; $r_{HSE} = 0.20$; $r_{UKBB} = 0.31$) and employment status with income ($r_{diff} = 0.10$; $r_{HSE} = -0.25$; $r_{UKBB} = -0.35$) (Fig. 2c). The application of probability weighting reduced bias induced by selective participation (median bias reduction, 0.97; mean, 0.91; range, 0.58–0.998). The estimates were very similar to the cross-validated model (median bias reduction, 0.96; mean, 0.90; range, 0.50–0.998), highlighting that overfitting was unlikely to be a problem.

Finally, Fig. 2d summarizes the changes in means and proportions following probability weighting, estimated for the auxiliary variables (in blue) as well as other UKBB variables (in orange) not used to construct the weights. Weighting resulted in a pseudo-sample with less favourable health outcomes and demographics, including more frequent mental illness (higher rates of schizophrenia and alcohol addiction) and poorer socio-economic status (higher deprivation index and lower job class).

In summary, using probability weighting, we created a pseudo-sample of the UKBB population achieving higher levels of representativeness along the 14 variables used in the weighting model. As a consequence, the weighting also changed the distributions of some variables not used in the weighting model (for example, an increased level of deprivation). Probability weighting thus provides a useful tool for examining bias due to selective participation in genomic studies, by evaluating how reweighting affects genome-wide results and downstream analyses.

### Probability-weighted GWA analyses

We next studied how the results of GWA analyses differ between weighted GWA (wGWA) ($n_{effective} = 94,643$ to $102,215$, depending on the trait) and standard GWA analyses ($\hat{\beta}$, $n = 263,464$ to $283,749$, depending on a trait). Reductions in the effective sample size in wGWA result from variability among the probability weights: when the weights are normalized to have a mean of one, the effective sample size simplifies to $n \times \{1/[\text{Var}(w_{in}) + 1]\}$. This quantity thus depends on the unweighted study sample size and on the variance of the normalized weights across study participants ($w_{in}$).

We assessed the impact of probability weighting on genome-wide findings in terms of changes in effect sizes across SNPs (contrasting weighted SNP effects, $\hat{\beta}_w$, to standard SNP effects, $\hat{\beta}$) and the number of significant SNP associations for 19 UKBB health-related traits collected at baseline (Fig. 3). First, Fig. 3a highlights the number of SNPs where weighting reduced ($(|\hat{\beta}| - |\hat{\beta}_w|)/|\hat{\beta}| \geq 0.2$) or increased ($(|\hat{\beta}| - |\hat{\beta}_w|)/|\hat{\beta}| \leq -0.2$) SNP effect sizes. Among all genome-wide hits (1,690, with $P < 5 \times 10^{-8}$), effect size reduction following weighting was

more common (420 SNPs, 24.85% of all genome-wide SNPs) than increase (290 SNPs, 17.16% of SNPs). More specifically, effect size increase was the most common for cancer (57% of SNPs), loneliness (50%), education (33%) and reaction time (33%), whereas reduction was present for depression/anxiety (67%), coffee intake (63%) and smoking status (58% of SNPs). While a shift towards more representativeness led to both effect size increases and decreases, we found no evidence of changes in the direction of effects (Supplementary Section 3.2).

Second, with respect to genome-wide discovery (Fig. 3b), we found that of all SNPs identified in either wGWA or GWA analyses ($n = 1,690$ across all phenotypes), 25 SNPs (1.48%) reached significance only in the weighted analyses. We found new SNPs for 12 of the 19 included traits, most notably for depression and anxiety (50% new genome-wide SNPs), cancer (29%) and loneliness (25%). The detailed results are listed in Supplementary Table 3 and plotted in Supplementary Figs. 2 and 3.

### Probability-weighted GWA analysis on sex

The UKBB included proportionally more women (female$_{UKBB}$ = 54.38%) than its target population (female$_{HSE}$ = 50.74%; female$_{CENSUS}$ = 50.62%). Probability weighting recovered the target population prevalence in the UKBB (weighted female$_{UKBB}$ = 50.36%). SNP heritability estimates ($h^2$) (Supplementary Fig. 4a) using wGWA led to almost half of that of the standard GWA ($h^2$ on liability scale, 1.2%, $P = 0.1$ in wGWA versus 2.1%, $P = 5.4 \times 10^{-11}$ in standard GWA). Supplementary Fig. 4b and Supplementary Table 4 display the SNP effects of 49 variants previously associated with sex ($P < 5 \times 10^{-8}$, in an independent sample of >2,400,000 volunteers) to estimates obtained from standard GWA and wGWA. Of those, 18 SNPs (36.73%) showed significantly lower sex-associated effects in wGWA. In contrast, only 3 SNPs (6.12%) had significantly lower sex-associated effects in standard GWA.

### GWA study on UKBB participation

We conducted a wGWA on UKBB participation in $n_{effective} = 102,215$ participants. A total of 28 SNPs reached genome-wide significance ($P < 5 \times 10^{-8}$), of which we selected 23 linkage disequilibrium (LD)-independent SNPs after clumping. Supplementary figures (Manhattan and QQ plots) and information (gene and phenotype annotation) for these SNPs are available in Supplementary Figs. 5 and 6 and Supplementary Tables 5 and 6.

SNP heritability for UKBB participation was $h^2 = 0.009$ (s.e. = 0.005; LD-score intercept, 1.055). LD-score regression analyses (Fig. 4b and Supplementary Table 7) implicated substantial genetic correlations between UKBB participation and phenotypes related to socio-economic factors and previously assessed participatory behaviour, including educational attainment ($r_g = 0.85$), income ($r_g = 0.77$), participation (provided e-mail address for recontact and mental health survey completion) ($r_g = 0.69$ and $r_g = 0.61$, respectively), intelligence ($r_g = 0.62$) and cigarette use (age of onset) ($r_g = -0.70$).

### Weighted SNP heritability and genetic correlation estimates

We next assessed differences in SNP heritability ($h^2_{DIFF} = h^2 - h^2_w$) and genetic correlations ($r_{g,DIFF} = r_g - r_{g,w}$) between standard GWA and wGWA analyses (Fig. 5). On average, heritability estimates differed by 1.5% (liability scale $|h^2_{DIFF}|$, 0.015; range, 0 to 0.05). $h^2_{DIFF}$ was the highest for BMI ($h^2 = 0.24$; $h^2_w = 0.19$), education ($h^2 = 0.21$; $h^2_w = 0.24$) and diabetes

---

**Fig. 2 | Performance of the UKBB probability weights. a**, Truncated (*) density curves of the normalized probability weights ($w_{in}$) for UKBB participants, ranging from 0.02 to 50.01. **b**, Standardized coefficients (and 95% confidence intervals) of variables predicting UKBB participation (HSE = 0; UKBB = 1) in univariate logistic regression models. Coefficients are provided for all UKBB participants and for males and females separately. **c**, Correlation coefficients among all auxiliary variables within the UKBB (obtained from weighted and unweighted analyses) and within the HSE. Highlighted in blue are results where the coefficients between the UKBB ($r_{UKBB}$) and the reference sample ($r_{HSE}$) deviated ($r_{diff} > 0.05$,

where $r_{diff} = |r_{HSE} - r_{UKBB}|$). **d**, Percentage change (for categorical variables) and change in means as a function of weighting, obtained for a number of health-related UKBB phenotypes, including the auxiliary variables (blue) and variables not used to construct the weights. Percentage change was estimated as the difference between the weighted ($p_w$) and unweighted proportion ($p$), divided by the unweighted value (($p_w - p$) / $p \times 100$). Change in means was expressed as a standardized mean difference, estimated as the difference between the unweighted mean ($m$) and the weighted mean ($m_w$), divided by the unweighted standard deviation ($m_w - m$/s.d.).

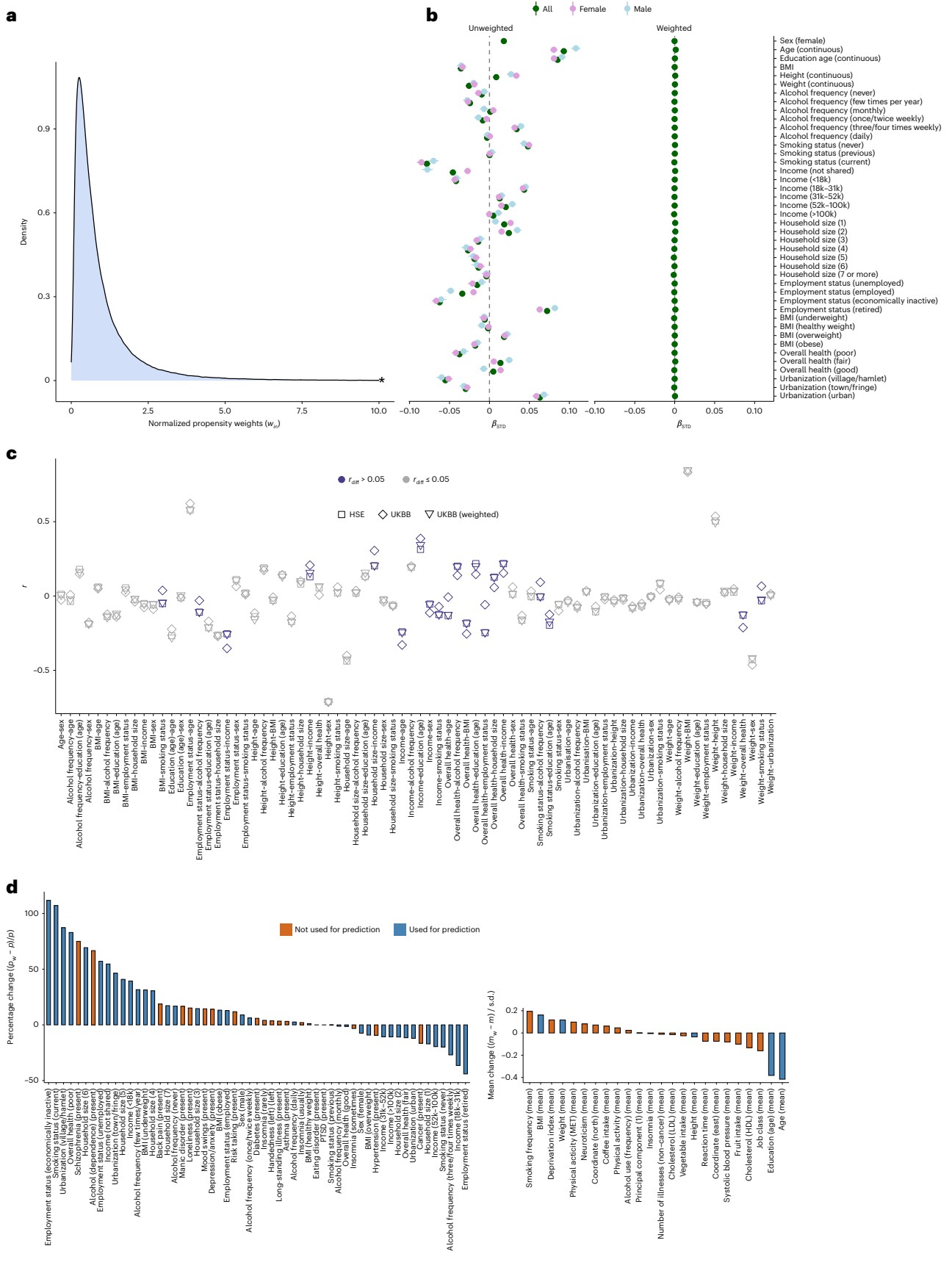

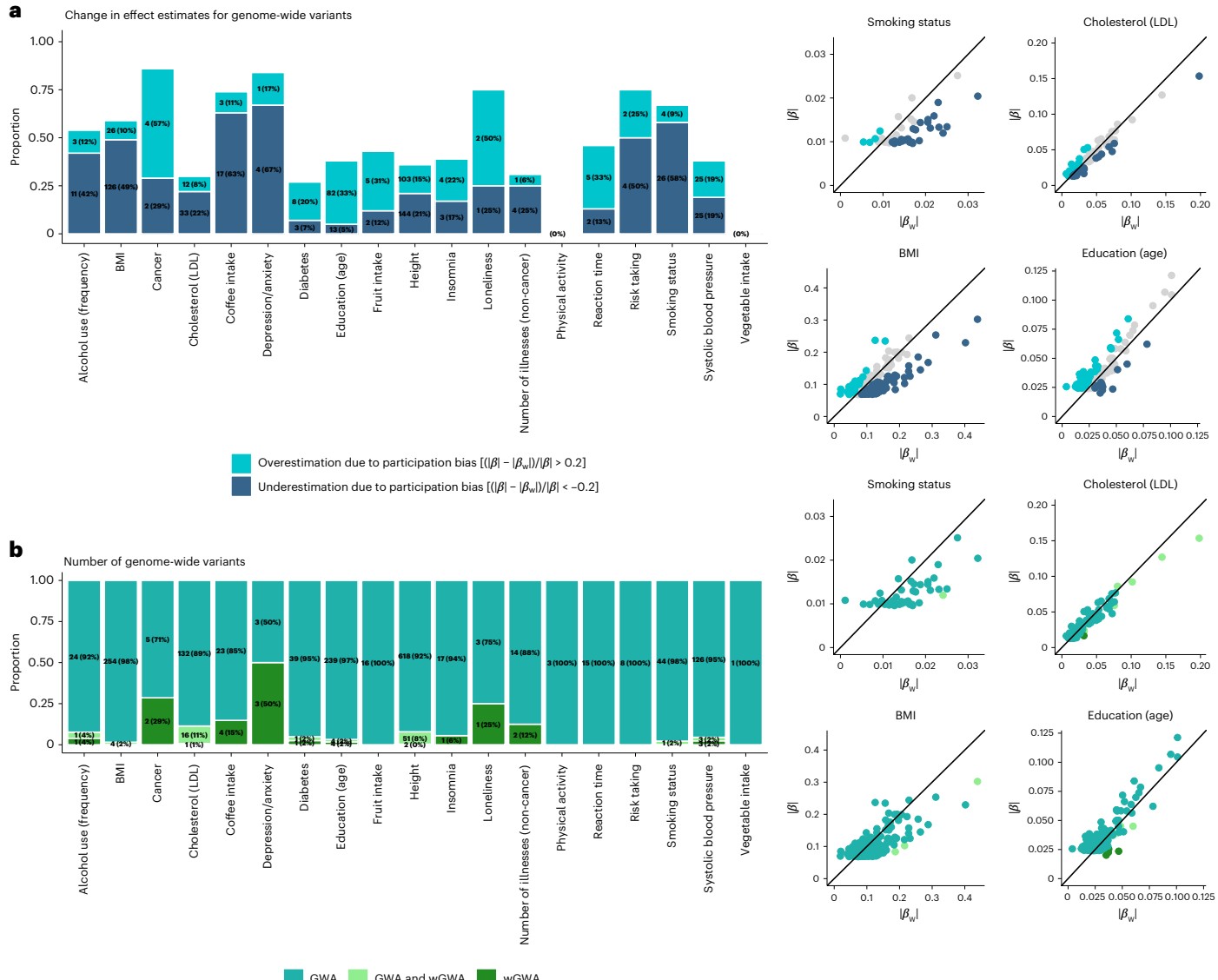

**Fig. 3 | SNP estimates from weighted and unweighted genome-wide analyses.** **a**,**b**, Summary of the comparison between SNP effects obtained from wGWA and standard GWA analyses on 19 traits. Panel **a** summarizes the proportions of overestimated and underestimated SNP effects as a result of participation bias. Shown in **b** are the numbers and proportions of SNPs reaching genome-wide significance in standard GWA, wGWA or both (GWA and wGWA). The scatter plots to the right plot the weighted ($|\beta_w|$) against the unweighted ($|\beta|$) SNP effects for four selected traits.

($h^2 = 0.19$; $h_w^2 = 0.16$). Of all assessed traits included in the LD-score regression ($n = 18$), five showed significant ($P_{FDR} < 0.05$) $h_{DIFF}^2$, of which four (80%) were lower and one (education) was higher in the more representative (weighted) sample. The weighted and unweighted heritability estimates are plotted in Supplementary Fig. 7, and additional statistics (for example, LD-score intercepts) are provided in Supplementary Table 8.

Concerning estimates of genetic correlations, we found an average difference of $|r_{g,DIFF}| = 0.07$ (range, 0 to 0.31) between results obtained from standard GWA and wGWA analyses. $r_g$ decreased the most notably for $r_g$(BMI, smoking status) ($r_g = 0.27$; $r_{g,w} = 0.13$), $r_g$(fruit intake, physical activity) ($r_g = 0.32$; $r_{g,w} = 0.18$) and $r_g$(alcohol use frequency, smoking status) ($r_g = 0.35$; $r_{g,w} = 0.21$). The increase in $r_g$ after weighting was the most prominent for $r_g$(insomnia, risk taking) ($r_g = 0.02$; $r_{g,w} = 0.31$), $r_g$(vegetable intake, physical activity) ($r_g = 0.3$; $r_{g,w} = 0.58$) and $r_g$(depression/anxiety, risk taking) ($r_g = 0.27$; $r_{g,w} = 0.47$). For five (3.27%) of the assessed trait pairs ($n = 153$) the weighted and standard genetic correlations were significantly ($P_{FDR} < 0.05$) different, of which education was the most implicated trait (Supplementary Fig. 8 and Supplementary Table 9). Change in the sign of genetic correlations because of participation bias was less common (17 of the 153 assessed trait pairs), but none of these $r_{g,DIFF}$ were significant ($P_{FDR} > 0.05$, Supplementary Section 3.3).

**Weighted MR estimates**

Figure 6 summarizes MR estimates with differences between the standard and weighted MR estimates ($\alpha_{DIFF} = \hat{\alpha} - \hat{\alpha}_w$).

On average, increasing sample representativeness led to an absolute change of 0.038 in standardized MR estimates (range, 0 to 0.15). Associations between lifestyle choices, including coffee intake on BMI ($\hat{\alpha} = 0.8$; $\hat{\alpha}_w = 0.65$), fruit consumption on LDL cholesterol ($\hat{\alpha} = 0.03$; $\hat{\alpha}_w = -0.12$) and fruit consumption on coffee intake ($\hat{\alpha} = 0.15$; $\hat{\alpha}_w = 0.01$) (Supplementary Fig. 9 and Supplementary Table 10), were the most affected. Of all exposure–outcome associations tested ($k = 234$), 14

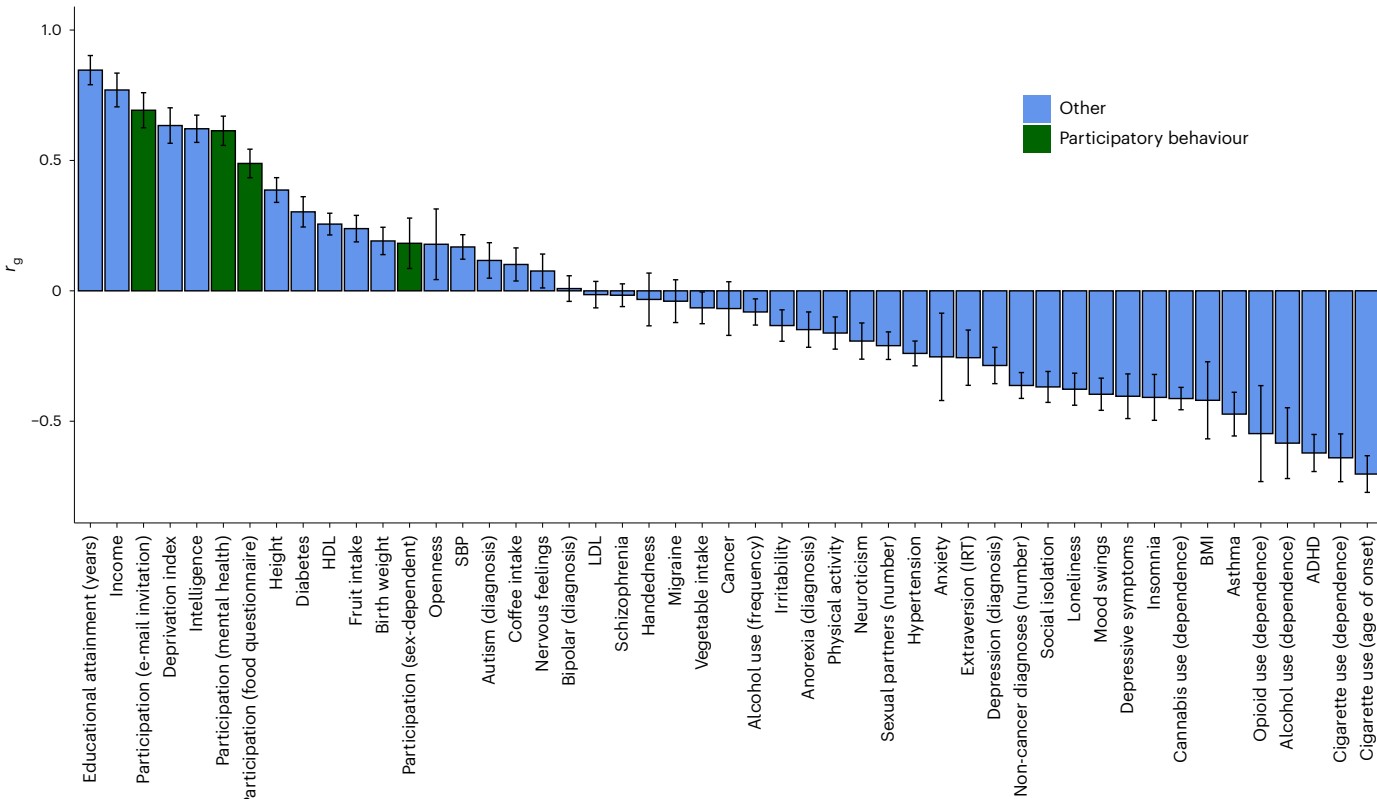

**Fig. 4 | GWA study on the liability to UKBB participation.** Shown are the genetic correlations ($r_g$) and corresponding 95% confidence intervals of UKBB participation (*n* standard GWA = 283,749) with traits indexing participatory behaviour (in green) and other traits (in blue) (including publically available summary statistics generated using standard GWA. SBP, systolic blood pressure; IR,: Item-response theory.

(6%) estimates were either decreased (2%, $|\hat{\alpha}| - |\hat{\alpha}_w| > 0.1$) or increased (4%, $|\hat{\alpha}| - |\hat{\alpha}_w| < -0.1$) after weighting. We found significant ($P_{FDR} < 0.05$) differential effects for two exposure–outcome associations (education on BMI and smoking status on fruit consumption). There was little evidence of changes in the direction of MR estimates as a result of weighting, which occurred for only two exposure–outcome pairs, neither of which was significant ($\alpha_{DIFF} P_{FDR} > 0.05$) (Supplementary Section 3.4).

## Discussion

While large volunteer-based biobanks are key to advancing genetic epidemiology, it is unclear to what extent selective participation impacts genotype–phenotype associations obtained from their data. In this work, we derived probability weights for the UKBB (based on 14 variables harmonized with data from a representative sample) and conducted inverse-probability-weighted GWA analyses on 19 traits. Conducting genome-wide analyses in a more representative (weighted) sample of the UKBB, we found that selective participation can distort genome-wide findings and downstream analyses.

Overall, increasing representativeness mostly affected the magnitude of effects rather than their direction. We found several differences in estimates in all sets of genome-wide analyses, in both directions (for example, a decrease in SNP effects after weighting for cancer and education and an increase in SNP effects for coffee intake and depression/anxiety). Of note, although effect size estimates can increase with the use of more representative samples, the increased standard errors of the inverse probability weighting (due to reduced effective sample size) make new SNP discovery difficult. Despite this caveat, using wGWA revealed new loci for 12 traits. Reweighting also changed heritability estimates, genetic correlations and MR estimates, most notably for socio-behavioural traits including education, diet, smoking and BMI.

In contrast, we observed smaller changes between wGWA and GWA estimates for molecular and physical traits (for example, low-density lipoproteins and systolic blood pressure). This pattern is in line with existing studies[23,24], as well as our findings of high genetic correlations between the liability to UKBB participation and socio-behavioural traits (particularly education, income and substance use). More broadly, different sources of bias probably affect similar phenotypes in genome-wide studies, in that genome-wide findings on socio-behavioural phenotypes are biased by selective participation[23,24], indirect genetic effects[3], assortative mating[4], error in measurements[25] and population stratification[26].

Our work builds on and extends recent efforts evaluating bias due to selective participation. We replicate findings showing that phenotypic exposure–outcome associations in the UKBB differ from those estimated in probability samples[13,15]: participation bias, defined as the difference in exposure–outcome associations in the UKBB and the reference sample (HSE), was substantial for several associations. For example, phenotypically, participation bias distorted the association of overall health with age and employment status. The application of probability weighting eliminated a significant proportion (>90%) of bias due to selective participation in the UKBB.

We highlight patterns of bias and point to areas of research that are the most impacted by this bias. Since GWA summary statistics are increasingly used in epidemiological research to study causal questions concerning education, diet and behaviour, greater care should be taken when relying on data obtained from non-random samples. If researchers cannot assess participation bias in biobank data (for example, in self-selected samples without a defined target population), their data may be of only limited use when scrutinizing genotype–phenotype relationships. As part of this work, we provide software to perform wGWA, which allows researchers to conduct sensitivity checks when

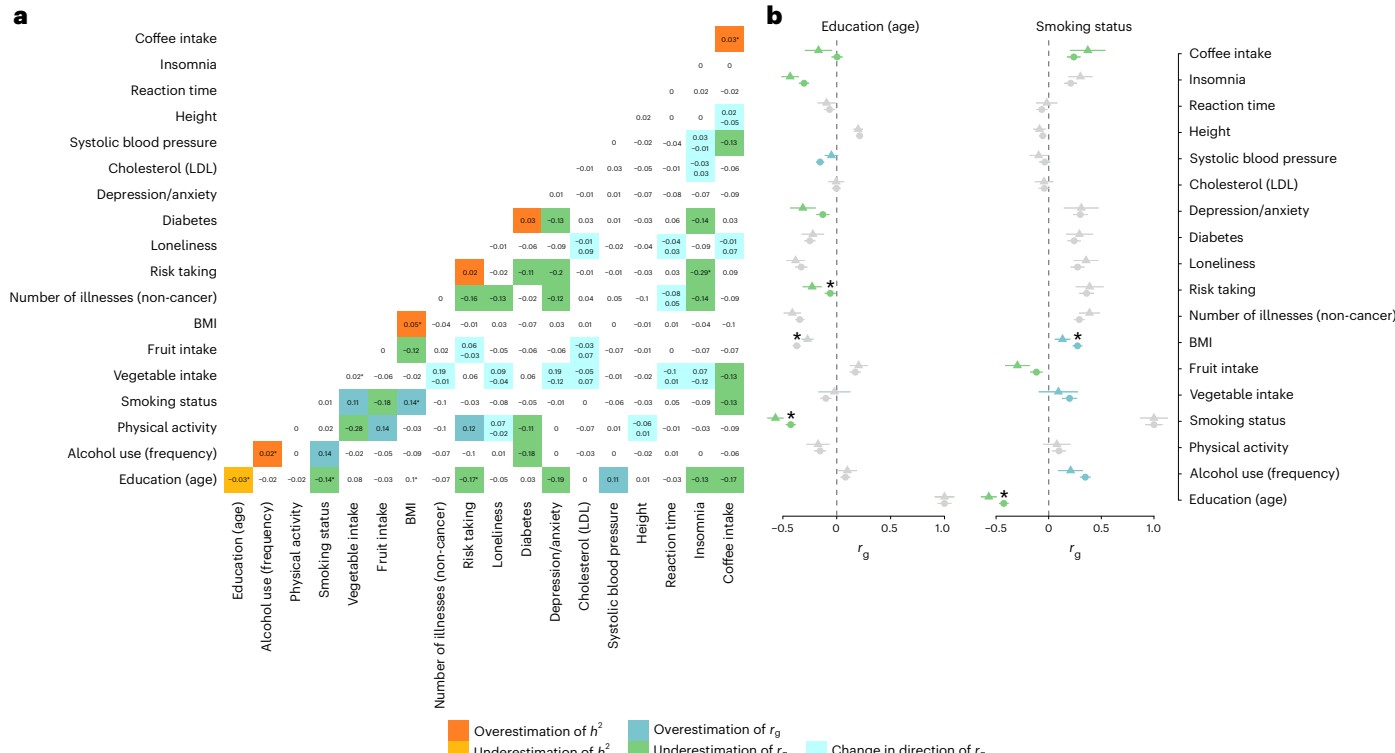

**Fig. 5 | Weighted SNP heritability and genetic correlation estimates.**
**a**, Differences in SNP heritability ($h^2_{\text{DIFF}} = h^2 - h^2_w$) and genetic correlations ($r_{g,\text{DIFF}} = |r_g| - |r_{g,w}|$) obtained from weighted and standard GWA analyses. The diagonal shows the differences in SNP heritability, where biases leading to overestimation ($h^2_{\text{DIFF}} > 0.02$) are plotted in orange and biases leading to underestimation ($h^2_{\text{DIFF}} < -0.02$) are plotted in yellow. The off-diagonal highlights overestimated genetic correlations ($r_{g,\text{DIFF}} > 0.1$) in blue and underestimated

genetic correlations ($r_{g,\text{DIFF}} < -0.1$) in green. Tiles coloured in turquoise index genetic correlations where $r_g$ and $r_{g,w}$ show opposite directions (with $r_g$ printed at the top and $r_{g,w}$ printed at the bottom of the tile). **b**, Estimates of genetic correlations ($r_g$ shown as circles; $r_{g,w}$ shown as triangles) and the corresponding 95% confidence intervals for two selected traits. The asterisks indicate estimates showing significant differences ($P_{\text{FDR}} < 0.05$). All $P$ values are from two-sided tests and are corrected for multiple testing using FDR correction (controlled at 5%).

relying on non-representative samples. Alternatively, recruitment schemes incorporating probability sampling can help reduce bias, but samples are typically small given the substantial costs associated with recruitment.

Our results should be interpreted with caution. First, while the application of probability weighting successfully reduced bias resulting from selective participation in the UKBB based on our 14 variables, residual bias still exists. We may have missed important factors independently predicting UKBB participation when modelling participation probability, as we chose our auxiliary variables on the basis of the availability of variables that could be harmonized between the UKBB and the reference sample. Still, some of these omitted variables may be proxied by (the combination of) some of the 14 variables, hence not compromising the probability weights. Probability weighing would not correct bias in situations where the exposure and the outcome of interest both link to an aspect of study participation that is unrelated to the auxiliary variables. This also means that wGWA for outcome traits such as education level is expected to be accurate, since this trait has been used when modelling participation probability. Finally, even for outcome traits completely unrelated to the 14 auxiliary variables but linked to traits influencing study participation, it is extremely unlikely that wGWA would be more biased than unweighted GWA. Hence, when substantial differences are observed between wGWA and standard GWA results, it is likely that the latter is (more) biased. Still, weighting—like any other method of adjusting for non-representativeness—should therefore be considered as only the second-best option when tackling participation bias, as only the implementation of probability sampling at the recruitment stage can ensure full elimination of this type of bias.

Second, when choosing a reference population, there is a trade-off between the representativeness of the reference sample and the number of available variables to match the samples. We chose to use the HSE as a reference sample to strike a balance between these two factors, but biases can remain if the reference sample is not representative enough. Third, genome-wide analyses were restricted to phenotypes with little missing data. This is a shortcoming since traits with substantial missing data are perfect candidates for characteristics influencing participation. We therefore did not evaluate the impact of participation bias on variables collected at follow-up.

Finally, the UKBB probability weights are sample-specific, constructed for a sample that is better educated, healthier and older and includes more women than the target population. Bias due to selective participation will differ across study contexts, and the participation mechanisms evaluated in this study are therefore not generalizable to other cohorts. For example, large health-registry-based biobanks, where older individuals with poorer health tend to be over-represented, do not have the healthy-volunteer bias but have different kinds of selection biases[27]. Similarly, the genome-wide results discussed here can be generalized only to adults of European genetic ancestry who also self-identify as white. Future work should also assess the impact of participation bias in more diverse samples, notably other ancestries and racial and ethnic groups, as well as younger individuals.

In conclusion, our results highlight that GWA and downstream analyses are sensitive to bias resulting from selective participation, most visibly for socio-behavioural traits. Moving forward, more efforts ensuring either sample representativeness or methods correcting for participation bias are paramount, especially when studying the genetic underpinnings of behaviour, lifestyles and educational outcomes.

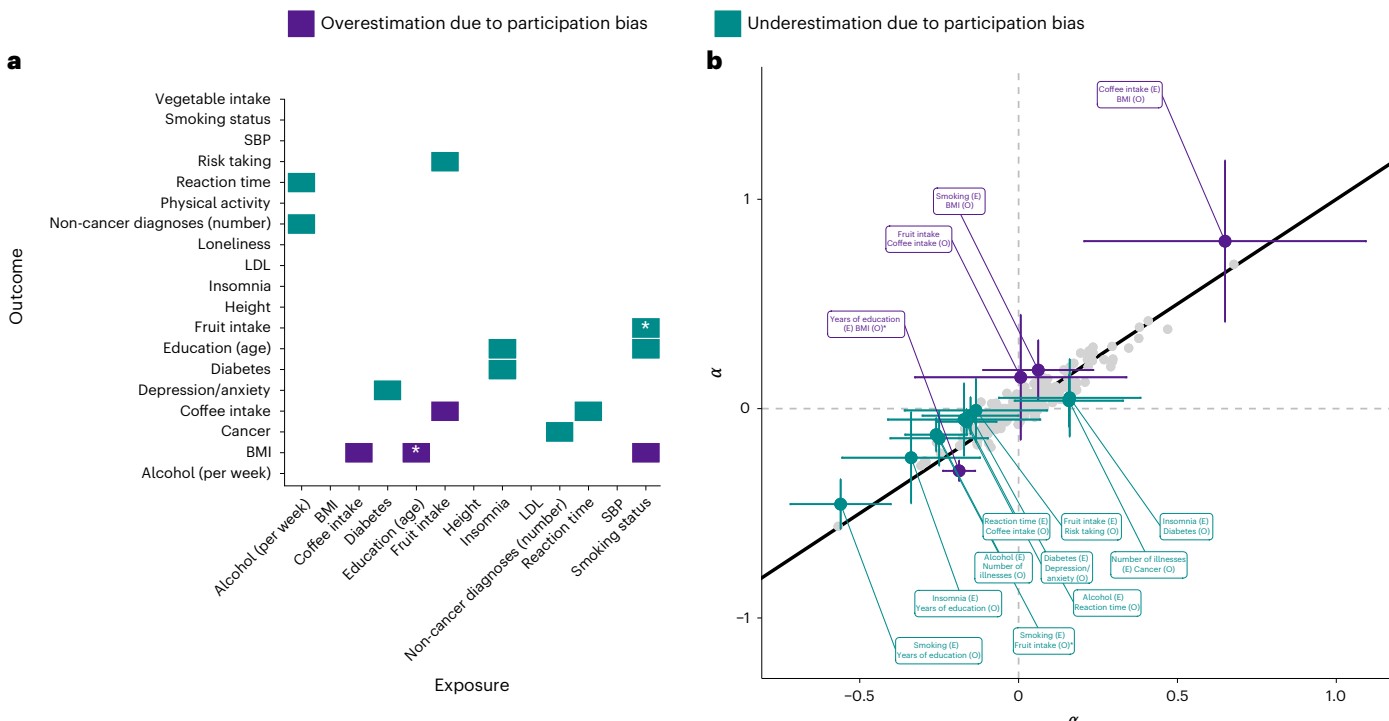

**Fig. 6 | Effect of participation bias on MR estimates of exposure–outcome associations. a,b,** Summary of results obtained from weighted ($\hat{\alpha}_w$) and standard ($\hat{\alpha}$) MR. MR estimates subject to overestimation ($|\hat{\alpha}| - |\hat{\alpha}_w| > 0.1$) as a result of participation bias are highlighted in violet. MR estimates subject to underestimation ($|\hat{\alpha}| - |\hat{\alpha}_w| < -0.1$) are highlighted in cyan. The asterisks highlight results where $\hat{\alpha}$ and $\hat{\alpha}_w$ showed significant ($P_{FDR} < 0.05$) differences. The error bars (**b**) indicate the 95% confidence intervals corresponding to $\hat{\alpha}$ and $\hat{\alpha}_w$. All $P$ values are from two-sided tests and are corrected for multiple testing using FDR correction (controlled at 5%).

## Methods

We first derived a model for participation probability by comparing 14 harmonized characteristics of the UKBB sample with those of a representative sample. Using the estimated participation probabilities, we conducted wGWA analyses on 19 UKBB traits. Second, to explore the genetic basis of UKBB participation, we conducted a GWA on the participation probability and evaluated the genetic findings. Finally, comparing wGWA results with those obtained from standard GWA analyses, we assessed the impact of participation bias on the estimation of three frequently studied quantities: (1) the effect of genetic markers on complex traits, (2) heritability and genetic correlation estimates, and (3) exposure–outcome associations obtained from MR.

### Samples

**UKBB.** The UKBB is a large-scale prospective population-based research resource focusing on the role of genetic, environmental and lifestyle factors in health outcomes in middle age and later life. More than 9,000,000 men and women between 40 and 69 registered with the UK National Health Service were invited to take part. Of those, 5.4% (~500,000 individuals) were recruited in 22 assessment centres across England, Wales and Scotland between 2006 and 2010[28,29]. Included in this study were data from UKBB participants of European genetic ancestry who also identify as white and passed standard GWA analysis quality control measures[30]. We further filtered the sample according to geographic region (excluding individuals from Scotland and Wales) to match the geographic regions included in the reference sample (HSE), and we removed individuals with missing data in the auxiliary variables used to generate the propensity scores (further described below). The UKBB resource was approved by the UKBB Research Ethics Committee, and all participants provided written informed consent to participate.

**HSE.** The HSE is an annual probability sample set out to measure health and related behaviours in a nationally representative sample of adults and children living in private households in England[31]. In our study, we included data from five cohorts recruiting a sample of more than 80,000 individuals between 2006 and 2010 (that is, the UKBB recruitment period). We applied the same inclusion criteria to the HSE data as used for UKBB recruitment, retaining only individuals aged between 40 and 69 years who self-identify as white. HSE response rates ranged between 64% and 68%[31]. HSE sample weights are supplied to account for the unequal probabilities of selection and non-response[32], weighing individuals as a function of sex, household type, region and social class. In this study, the HSE weights were incorporated in LASSO regression predicting UKBB participation (described below).

**UK Census data.** We also exploited data from the 2011 Census Microdata, a 5% sample of anonymized individual-level Census records[33], which runs every ten years to collect basic demographic variables (for example, educational attainment, age and general health) through a paper-based or online questionnaire. With a 95% response rate, the UK Census Microdata is highly representative of the UK population. We applied the same selection criteria to the Census data as to the UKBB and HSE (that is, filtered according to geographic region, ethnic group and age), resulting in a relevant sample of $n = 895,649$. We extracted all variables that could be harmonized with the UKBB and HSE data (further described in the Supplementary Information). The Census data were solely used to assess the level of representativeness of the HSE, by comparing the distributions and associations between variables present in both the HSE and the Census sample. For the generation of UKBB probability weights, we used the HSE sample, given its richer phenotypic data, which are critical for accurate weight estimation.

## Analysis

**Auxiliary variables.** We adjusted for participation bias in the UKBB using probability weighting[34]. This approach adjusts for non-response bias by weighting over-represented and under-represented individuals, thereby creating a pseudo-population that is more representative of its target population[35]. Probability weighting relies on auxiliary variables available for both a selected (non-representative) and a representative reference sample. In this study, we selected auxiliary variables tapping into dimensions related to health, lifestyle, education and basic demographics. We included all variables that could be harmonized across the two datasets (HSE and UKBB) with few missing observations (that is, <50,000 in the UKBB and <500 in the HSE). Fourteen variables derived from 12 measures were included and harmonized across the two datasets. The five continuous variables included age, BMI, weight, height and education (age when the individual completed full-time education). The nine categorical variables included household size (1, 2, 3, 4, 5, 6, or 7 or more), sex (male or female), alcohol consumption frequency (never, a few times per year, monthly, once or twice weekly, three or four times weekly, or daily), smoking status (never, previous or current), employment status (employed, economically inactive, retired or unemployed), income (<18k, 18k–31k, 31k–52k, 52k–100k or >100k), obesity status (underweight, healthy weight, overweight or obese), overall health (poor, fair or good) and degree of urbanisation (village/hamlet, town/fringe, urban). Further details of the coding of the variables in each dataset are provided in the Supplementary Information.

**Construction and evaluation of UKBB probability weights.** To derive the model for participation probability, we first combined the harmonized UKBB data with the data from the reference sample (HSE). We then used LASSO regression in glmnet[36] to predict UKBB participation ($P_i$, with UKBB = 1; HSE = 0), conditional on the harmonized auxiliary variables described above. We included 14 main effects (5 continuous variables and 9 binary/categorical variables) in the model. All categorical and binary variables were entered as dummy variables, indexing each possible level of the variable. In addition, we included all possible two-way interaction terms among the dummy and continuous variables, resulting in 903 included predictors. LASSO performs variable selection by shrinking the coefficients for variables that contribute the least to prediction accuracy. The shrinkage is controlled by the tuning parameter ($\lambda$), which was obtained using fivefold cross-validation that minimizes the cross-validated error.

The predicted probabilities ($P_i$) were then used to build the individual sampling weights ($w_i$). The weights were calculated as an extension of standard inverse probability weights ($w_i = (1 - P_i)/P_i$), designed to make the weighted sample estimates conform to the population estimates[35]. To assess the performance of the generated weights, we evaluated the extent to which the weighting recovered means (for continuous variables) and prevalences (for binary traits) in the UKBB and hence mitigated participation bias. We also quantified participation bias as the differences between the correlations among all auxiliary variables within the UKBB ($r_{UKBB}$) and the HSE ($r_{HSE}$). The degree to which the weighted correlations ($r_{UKBB,w}$) reduced bias was estimated as ($|r_{HSE} - r_{UKBB}| - |r_{HSE} - r_{UKBB,w}|)/(|r_{HSE} - r_{UKBB}|)$, where a value of one indicates that weighting fully eliminated bias. The weighted means (and proportions) for a given variable ($X_i$) were estimated using the weights ($w_i$), with the expression $\frac{1}{W} \sum_{i=1}^{N} w_i X_i$, where $W = \sum_{i=1}^{N} w_i$.

We further evaluated whether overfitting was a problem by rerunning LASSO in train–test splits of the data (fivefold leave-one-out cross-validation, with a split ratio of 80:20). Here we used the training sample (80% of the data) for model estimation and the test sample (20% of the data) to generate the out-of-sample predicted probabilities. The degree of participation bias reduction was then compared between the out-of-sample predicted probabilities and the full-sample probabilities.

**Probability-weighted GWA analyses.** To evaluate the extent to which SNP effects were distorted by participation bias in the UKBB, we conducted wGWA analyses. wGWA was performed for 19 UKBB health-related traits collected at baseline with few missing observations ($n_{missing} < 50,000$). Some of these traits (education, frequency of alcohol use, weight, height and smoking status) were used in the model deriving the probability weights. The coding of all variables, genotyping, imputation and quality control procedures are described in the Supplementary Information. Additional quality control filters for genome-wide analyses were applied to select participants (that is, restricting the sample to unrelated individuals of European genetic ancestry and excluding individuals with high missing rate and high heterozygosity on autosomes) and genetic variants (Hardy–Weinberg disequilibrium $P > 1 \times 10^{-6}$, minor allele frequency > 1% and call rate > 90%).

We obtained unweighted SNP estimates ($\hat{\beta}$) from a standard ordinary least squares linear regression model. The weighted SNP estimates ($\hat{\beta}_w$) were obtained from weighted least squares regression. All GWA analyses were conducted in LDAK (version 5.2)[37,38], which was extended to accommodate sampling weights in a linear weighted least squares model (linear; sample-weights). The standard least squares estimate of the variance is based on the assumption of homoskedasticity (that is, that the residual variance is constant across individuals). Since the use of sampling weights violates this assumption, we used the Huber–White estimator[39] to estimate the variance of the coefficients:

$$\widehat{\beta_w} = (\mathbf{X}'W\mathbf{X})^{-1}(\mathbf{X}'W\mathbf{Y})$$

$$\mathrm{Var}\left(\widehat{\beta_w}\right) = (\mathbf{X}'W\mathbf{X})^{-1}(\mathbf{X}'WDW\mathbf{X})(\mathbf{X}'W\mathbf{X})^{-1}$$

with

$$D = \mathrm{diag}\left[\left(\mathbf{Y} - \mathbf{X}\widehat{\beta_w}\right)^2\right]$$

where **Y** denotes the phenotypic outcome vector, $W$ is a diagonal matrix with the probability weights sitting on the diagonal and **X** is a column vector of the genotype values.

Both models included the same covariates (PC1–PC5, sex, age and batch effect). We applied a linear model to all outcomes (continuous and binary traits). This was done to allow for the standardization of SNP estimates and to ensure the comparability of effect sizes. A more detailed discussion on the advantages and disadvantages of using a linear over a logistic model for binary outcomes is provided by von Hippel[40,41], as well as the Neale Lab[42] discussing its application specifically when using UKBB data.

Two additional sets of analyses were conducted to explore the genetic basis of UKBB participation. First, we conducted autosomal wGWA and standard GWA on biological sex and evaluated whether wGWA reduced sex-differential participation bias. As previously suggested[23], autosomal heritability linked to biological sex could result from sex-differential participation. As such, reduced heritability estimates in wGWA compared with GWA would provide evidence for the utility of wGWA for participation bias correction. In addition, we compared the resulting SNP effects with the effects of previously identified sex-associated variants ($P < 5 \times 10^{-8}$). Here 49 variants assessed in an independent sample of >2,400,000 volunteers curated by 23andMe[23] were selected.

Second, we conducted a genome-wide analysis on the liability to UKBB participation, by including the individual participation probabilities as the outcome of interest in wGWA. The application of standard GWA analysis is not possible in this context, as this approach stratifies for the outcome of interest by selecting a subset of the population willing to participate. LD-independent SNPs reaching genome-wide

significance ($P < 5 \times 10^{-8}$) were selected via clumping (clump-kb, 250; clump-r2, 0.1; following standard recommendations[43]). PhenoScanner[44], a database of genotype–phenotype associations from existing GWA studies, was used to explore previously identified associations of lead SNPs with other phenotypes. Genetic correlations with other traits were estimated using LD-score regression[45] as implemented in the R package GenomicSEM[46]. The summary statistic files used in LD-score regression were obtained for 49 health and behavioural phenotypes, using publicly available summary statistic files accessible via consortia websites or the MRC-IEU OpenGWAS project (https://gwas.mrcieu.ac.uk)[47] (see Supplementary Table 11 for the details).

**LD score regression and heritability estimates.** SNP heritability estimates were obtained for both the standard GWA and wGWA output ($h^2$ and $h_w^2$, respectively) using LD score regression as implemented in GenomicSEM. We applied the default settings (restricted SNPs to minor allele frequency > 0.01, LD scores from the European-ancestry sample in the 1000 Genomes Project[48]). For binary phenotypes, the observed scale was converted to the liability scale[49], where the population prevalence was set to be equal to the weighted prevalence in the UKBB. We also estimated bivariate genetic correlations among all phenotypes included in standard GWA and wGWA ($r_g$ and $r_{g,w}$, respectively). To compare the estimates obtained from wGWA and standard GWA, we calculated the difference ($r_{g,DIFF} = r_g - r_{g,w}$ and $h_{DIFF}^2 = h^2 - h_w^2$) and used the following test statistic (here exemplified for $r_{g,DIFF}$):

$$Z_{r_g} = \frac{r_{g,DIFF}}{s.e.(r_{g,DIFF})}$$

$$s.e.(r_{g,DIFF}) = \sqrt{s.e.(r_g)^2 + s.e.(r_{g,w})^2 - 2r \; s.e.(r_g) \; s.e.(r_{g,w})}$$

The correlation coefficients $r(h^2, h_w^2)$ and $r(r_g, r_{g,w})$ were obtained from 200-block jackknife analysis. For this, we split the genome into 200 equal blocks of SNPs and removed one block at a time to perform jackknife estimation.

**MR analyses.** To evaluate the impact of selection bias when using MR, we assessed whether sample weighting altered MR estimates. As genetic instruments, we selected LD-independent (clump-kb, 10,000; clump-r2, 0.001; adhering to standard MR protocols[50]) SNPs reaching genome-wide significance ($P < 5 \times 10^{-8}$) in either wGWA or standard GWA for a given phenotype. Phenotypes with few (<10) genetic instruments were not included in the MR analyses. We used the inverse-variance weighted (IVW) MR estimator, which combines the ratio estimates of the individual genetic variants $G_j$ to derive the causal effect ($\hat{\alpha}_{IVW}$). The ratio estimate is $\hat{\alpha}_j = \hat{\beta}_j^{OUT}/\hat{\beta}_j^{EXP}$, where $\hat{\beta}_j^{EXP}$ corresponds to the SNP–exposure association and $\hat{\beta}_j^{OUT}$ corresponds to the SNP–outcome association. Since the IVW estimator assumes that the uncertainty in the genetic association with the exposure is zero, we used the following correction[51] to account for selected genetic variants ($\hat{\beta}_j^{EXP}$) that were genome-wide significant in one analysis (for example, standard GWA) but not the other (for example, wGWA) for the same trait: $\hat{\alpha}_{IVW,corrected} = \hat{\alpha}_{IVW}\frac{S^2}{\hat{\sigma}}$, where $S^2 = \frac{1}{m-1}\sum_{j=1}^{m}\left(\hat{\beta}_j^{EXP} - \overline{\hat{\beta}_{EXP}}\right)^2$ and $\hat{\sigma}^2 = S^2 - \frac{1}{m}\sum_{j=1}^{m} Var\left(\hat{\beta}_j^{EXP}\right)$, where $m$ refers to the number of SNPs selected as instruments. The corresponding variance was estimated as $Var(\hat{\alpha}_{IVW,corrected}) = Var(\hat{\alpha}_{IVW})\frac{S^2}{\hat{\sigma}^2}$.

For each exposure–outcome association, we obtained (1) an MR estimate using the SNP effects from standard GWA analyses and (2) an MR estimate using the SNP effects from wGWA analyses. We included in MR the standardized SNP effects and standard errors (that is, the effect of the genotype on the standardized outcome), which were derived using the following formula[52]: $\beta_{STD} = Z/\sqrt{2p(1-p)(n+Z^2)}$ and

$s.e.(\beta_{STD}) = 1/\sqrt{2p(1-p)(n+Z^2)}$, where $n$ is the sample size, $p$ is the minor allele frequency and $Z$ is the SNP effect $\hat{\beta}$ divided by its standard error ($Z = \hat{\beta}/s.e.(\hat{\beta})$). Of note, when standardizing the weighted estimates ($\hat{\beta}_w$), $n$ was replaced by the effective sample size ($n_{effective} = W^2/\sum_{i=1}^{N} w_{in}^2$) to account for the unequal contribution per observation. $w_{in}$ refers to the normalized probability weights, obtained by dividing $w_i$ by its mean ($w_{in} = w_i/\overline{w_i}$).

To compare the standard ($\hat{\alpha}$) to the weighted MR ($\hat{\alpha}_w$) estimates, we estimated $\alpha_{DIFF}$ ($\hat{\alpha} - \hat{\alpha}_w$) and the corresponding test statistic as $Z = \alpha_{DIFF}/s.e.(\alpha_{DIFF})$, where

$$s.e.(\alpha_{DIFF}) = \sqrt{s.e.(\hat{\alpha})^2 + s.e.(\hat{\alpha}_w)^2 - 2r \; s.e.(\hat{\alpha}) \; s.e.(\hat{\alpha}_w)}.$$

The correlation coefficient was derived using a jackknife procedure, where we performed MR leaving out each SNP in turn to then calculate the correlation $r(\hat{\alpha}, \hat{\alpha}_w)$ from these results. The results were corrected for multiple testing using FDR correction (controlled at 5%), correcting for the total number of conducted MR analyses.

### Reporting summary

Further information on research design is available in the Nature Portfolio Reporting Summary linked to this article.

### Data availability

All summary statistic files generated using standard and weighted genome-wide analyses are accessible on the GWAS catalogue (https://www.ebi.ac.uk/gwas/) at the accession numbers GCST90267266 to GCST90267307. The UKBB probability weights generated as part of this study are available via the UK Biobank repositories.

### Code availability

The following software was used to run the analyses: LDAK (http://dougspeed.com/downloads/; a tutorial on how to perform standard and weighted genome-wide analyses is available at https://tabeaschoeler.github.io/TS2021_UKBBweighting/wGWA.html), TwoSampleMR (https://mrcieu.github.io/TwoSampleMR/) and GenomicSEM (https://github.com/GenomicSEM/GenomicSEM). All analytical scripts are available at https://github.com/TabeaSchoeler/TS2021_UKBBweighting.

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

## Acknowledgements

This research has been conducted with the UK Biobank Resource under application number 16389; we thank all biobank participants for sharing their data. We thank all participants involved in the Health Survey England and the 2011 Census Microdata, and we thank the Office for National Statistics for granting access to the data. This study would not have possible without the use of publicly available genome-wide summary data and software tools. We acknowledge these resources and thank the research participants, research teams and institutions that have contributed to this research. The computations were performed on the HPC cluster of the Lausanne University Hospital. We thank Y. Tillé for helpful discussions and relevant comments. Z.K. was funded by the Swiss National Science Foundation (grant no. 310030-189147). T.S. is funded by a Wellcome Trust Sir Henry Wellcome fellowship (grant no. 218641/Z/19/Z). For the purpose of open access, we have applied a CC BY public copyright licence to any Author Accepted Manuscript version arising from this submission. J.-B.P. has received funding from the European Research Council under the European Union's Horizon 2020 research and innovation programme (grant agreement no. 863981) and is supported by the Medical Research Foundation 2018 Emerging Leaders 1st Prize in Adolescent Mental Health (grant no. MRF-160-0002-ELP-PINGA). D.S. is supported by the Aarhus University Research Foundation, by the Independent Research Fund Denmark under project no. 7025-00094B and by a Lundbeck Foundation Experiment Grant. The funders had no role in study design, data collection and analysis, decision to publish or preparation of the manuscript.

## Author contributions

Z.K. and T.S. conceptualized the study. T.S. performed the statistical analyses. D.S. provided the software. Z.K., D.S., J.-B.P., E.P. and N.P. discussed the results and provided comments on the paper. All authors critically reviewed the manuscript.

## Funding

## Competing interests

The authors declare no competing interests.

## Additional information

**Correspondence and requests for materials** should be addressed to Tabea Schoeler or Zoltán Kutalik.

# Reporting Summary

## Statistics

For all statistical analyses, confirm that the following items are present in the figure legend, table legend, main text, or Methods section.

| n/a | Confirmed | |
|---|---|---|
| ☐ | ☒ | The exact sample size (*n*) for each experimental group/condition, given as a discrete number and unit of measurement |
| ☐ | ☒ | A statement on whether measurements were taken from distinct samples or whether the same sample was measured repeatedly |
| ☐ | ☒ | The statistical test(s) used AND whether they are one- or two-sided<br>*Only common tests should be described solely by name; describe more complex techniques in the Methods section.* |
| ☐ | ☒ | A description of all covariates tested |
| ☐ | ☒ | A description of any assumptions or corrections, such as tests of normality and adjustment for multiple comparisons |
| ☐ | ☒ | A full description of the statistical parameters including central tendency (e.g. means) or other basic estimates (e.g. regression coefficient) AND variation (e.g. standard deviation) or associated estimates of uncertainty (e.g. confidence intervals) |
| ☐ | ☒ | For null hypothesis testing, the test statistic (e.g. *F*, *t*, *r*) with confidence intervals, effect sizes, degrees of freedom and *P* value noted<br>*Give P values as exact values whenever suitable.* |
| ☒ | ☐ | For Bayesian analysis, information on the choice of priors and Markov chain Monte Carlo settings |
| ☒ | ☐ | For hierarchical and complex designs, identification of the appropriate level for tests and full reporting of outcomes |
| ☒ | ☐ | Estimates of effect sizes (e.g. Cohen's *d*, Pearson's *r*), indicating how they were calculated |

*Our web collection on statistics for biologists contains articles on many of the points above.*

## Software and code

Policy information about availability of computer code

| Data collection | This research has been conducted with the UK Biobank Resource under application number 16389. |
|---|---|
| Data analysis | We used the following software to conduct the analyses: LD Score Regression as implemented in GenomicSEM (https://github.com/GenomicSEM/GenomicSEM), LDAK (http://dougspeed.com/downloads/), TwoSampleMR (https://mrcieu.github.io/TwoSampleMR/). All analytical scripts are available at https://github.com/TabeaSchoeler/TS2021_UKBBweighting |

For manuscripts utilizing custom algorithms or software that are central to the research but not yet described in published literature, software must be made available to editors and reviewers. We strongly encourage code deposition in a community repository (e.g. GitHub). See the Nature Portfolio guidelines for submitting code & software for further information.

## Data

Policy information about availability of data

All manuscripts must include a data availability statement. This statement should provide the following information, where applicable:

- Accession codes, unique identifiers, or web links for publicly available datasets
- A description of any restrictions on data availability
- For clinical datasets or third party data, please ensure that the statement adheres to our policy

Data availability: Standard and probability weighted UK Biobank association statistics, computed using LDAK version 5.2, will be made available through the GWAS catalog.

## Human research participants

Policy information about studies involving human research participants and Sex and Gender in Research.

| | |
|---|---|
| Reporting on sex and gender | We used self-reported sex (biological attribute) in our study. |
| Population characteristics | In genome-wide analyses, we included UK Biobank participants of European ancestry passing standard GWA analysis quality control measures. All analyses were adjusted for batch, principal components (PC1-PC5), age and sex. Exclusions during QC process (phenotypic and genetic) are detailed in the Methods. Demographic information about the sample is provided in Supplementary Table 3. |
| Recruitment | The UK Biobank (UKBB) is a prospective population-based research resource focusing on the role of genetic, environmental and lifestyle factors in health outcomes in middle age and later life. More than 9,000,000 men and women between 40 and 69 registered with the UK NHS were invited to take part. Of those, 5.4% (~500,000 individuals) were recruited in 22 assessment centres across England, Wales and Scotland between 2006 and 2010. |
| Ethics oversight | The UK Biobank resource was approved by the UK Biobank Research Ethics Committee and all participants provided written informed consent to participate. |

Note that full information on the approval of the study protocol must also be provided in the manuscript.

# Field-specific reporting

Please select the one below that is the best fit for your research. If you are not sure, read the appropriate sections before making your selection.

☒ Life sciences   ☐ Behavioural & social sciences   ☐ Ecological, evolutionary & environmental sciences

For a reference copy of the document with all sections, see nature.com/documents/nr-reporting-summary-flat.pdf

# Life sciences study design

All studies must disclose on these points even when the disclosure is negative.

| | |
|---|---|
| Sample size | We conducted inverse probability weighted genome-wide association analyses (Neffective=94,643 – 102,215) and standard GWA (N=263,464 – 283,749) in UKBB participants selected for genome-wide analyses (UK Biobank participants of European ancestry passing standard GWA analysis quality control measures).<br>Quality control filters for genome-wide analyses were applied to select participants (i.e., exclusion of related individuals, exclusion of non-White British ancestry based on principal components, high missing rate and high heterozygosity on autosomes) and genetic variants (Hardy–Weinberg disequilibrium P>1×10−6, minor allele frequency>1% and call rate>90%). |
| Data exclusions | We filtered the sample according to geographical region (excluding individuals from Scotland and Wales) to match the geographical regions included in the reference sample (HSE), and removed individuals with missing data in auxiliary variables used to generate the propensity scores. |
| Replication | We used the UK Biobank as it is the currently largest sample where participation bias correction through inverse weighted genome-wide association analyses can be performed. Our findings replicate previous genome-wide findings and highlight the extend to which these findings may be biased by selective participation. We did not select an independent replication sample. There are no genotype datasets of similar size in the UK for which sampling weights could be computed, making replication currently not feasible. |
| Randomization | Not applicable |
| Blinding | Not applicable |

# Reporting for specific materials, systems and methods

We require information from authors about some types of materials, experimental systems and methods used in many studies. Here, indicate whether each material, system or method listed is relevant to your study. If you are not sure if a list item applies to your research, read the appropriate section before selecting a response.

## Materials & experimental systems

| n/a | Involved in the study |
|-----|------------------------|
| ☒ | Antibodies |
| ☒ | Eukaryotic cell lines |
| ☒ | Palaeontology and archaeology |
| ☒ | Animals and other organisms |
| ☒ | Clinical data |
| ☒ | Dual use research of concern |

## Methods

| n/a | Involved in the study |
|-----|------------------------|
| ☒ | ChIP-seq |
| ☒ | Flow cytometry |
| ☒ | MRI-based neuroimaging |

