## [Peer Review File · Nature Human Behaviour]

Peer Review Information

Journal: Nature Human Behaviour

Manuscript Title: Participation bias in the UK Biobank distorts genetic associations and downstream analyses

Corresponding author name(s): Tabea Schoeler and Zoltán Kutalik

Reviewer Comments & Decisions: .

Decision Letter, initial version:

1st November 2022

Dear Dr Schoeler,

Thank you once again for your manuscript, entitled "Correction for participation bias in the UK Biobank reveals non-negligible impact on genetic associations and downstream analyses," and for your patience during the peer review process.

Your manuscript has now been evaluated by 3 reviewers, whose comments are included at the end of this letter. Although the reviewers find your work to be of interest, they also raise some important concerns. We are interested in the possibility of publishing your study in Nature Human Behaviour, but would like to consider your response to these concerns in the form of a revised manuscript before we make a decision on publication.

In your revision, we ask that you:

- 1) Please fully address Reviewer #1's concerns about the robustness of your GWAS of liability to UKBB participation
- 2) Reviewers point to an importance of the precise usages of terms relating to race/ethnicity. Please correct the usage of "ancestry" and "race/ethnicity";
- 3) In your comparisons, please make sure that you use phenotype measures that are comparable to those of other published GWASs;
- 4) Revise the framing to transparently discuss the limitations of your approach;
- 5) Please reframe your work as a sensitivity analysis for selection bias, acknowledge that you are unable to verify the absolute truth of your results based on the observed data, and remove all description of the UKBB sample as representative

In sum, we invite you to revise your manuscript taking into account all reviewer and editor comments. We are committed to providing a fair and constructive peer-review process. Do not hesitate to contact us if there are specific requests from the reviewers that you believe are technically impossible or unlikely to yield a meaningful outcome.

We hope to receive your revised manuscript within two months. I would be grateful if you could contact us as soon as possible if you foresee difficulties with meeting this target resubmission date.

- Include a "Response to the editors and reviewers" document detailing, point-by-point, how you addressed each editor and referee comment. If no action was taken to address a point, you must provide a compelling argument. When formatting this document, please respond to each reviewer comment individually, including the full text of the reviewer comment verbatim followed by your response to the individual point. This response will be used by the editors to evaluate your revision and sent back to the reviewers along with the revised manuscript.
- Highlight all changes made to your manuscript or provide us with a version that tracks changes.

[REDACTED]

We look forward to seeing the revised manuscript and thank you for the opportunity to review your work. Please do not hesitate to contact me if you have any questions or would like to discuss these revisions further.

Sincerely,

Arunas Radzvilavicius, PhD
Editor, Nature Human Behaviour
Nature Research

Reviewer expertise:

Reviewer #1: epidemiology, genetics, statistics

Reviewer #2: genomics, population health, stratification

Reviewer #3: epidemiology, medical statistics

REVIEWER COMMENTS:

Reviewer #1:

Remarks to the Author:

- It would be valuable to mention that, yes, epidemiological studies are typically subjected to healthy-volunteer bias, but that cannot be said for many hospital-based biobanks which are common, especially in US (but also FinnGen). The biobanks are more likely to enroll sicker individuals.

- It would be valuable to clarify which variables are self-reported vs objectively measured. For example in HSE was BMI self-reported and was this compared to measured BMI in UK Biobank?

- Because you use LDK software, wouldn't make more sense to calculate h^2 based on LDK heritability model assumptions? Or at least using stratified LD-score to get more accurate estimates?

- Figure 1, panel A: A little bit misleading that the distribution is truncated, even if mentioned in the legend. Can you add a symbol to clearly indicate truncation?
- Can the authors comment more extensively about the smaller effective sample size due to inverse probability weighting? Is this reduction function of the sample size of the representative study used to derive weights? Or what determines the reduction in effective sample size? This is a rather important consideration if weighted GWAS should be implemented.
- Related to the comment above. Are the 1.48% new SNPs from wGWAS identified despite the reduction in effective sample size?
- The Manhattan plot for the GWAS of the liability to UKBB participation doesn't look very robust. Did you filter out rare variants? For example, rs189384883 does not seem to have LD partners with similar statistical significance. Many of the signals do not have the traditional "GWAS peak". I would be careful about interpreting individual SNPs obtained from this analysis.

Reviewer #2:

Remarks to the Author:

Below is my review of manuscript NATHUMBEHAV-22092529 titled, "Correction for participation bias in the UK Biobank reveals non-negligible impact on genetic associations and downstream analyses." This study seeks to investigate if the use of inverse-probability weights from a representative sample (Health Survey England) applied to a non-representative volunteer-based sample (the UK Biobank) can improve estimates obtained from three types of genomic analyses (GWAS, LD score regression, Mendelian Randomization). Importantly, this study addresses ongoing weakness in both genetic discovery and the application of those discoveries via PGSs and other methods in population-level research.

Major comments:

1. The author(s) need to better clarify the distinction between race/ethnicity and genetic ancestry and when/how they use both to define their analytic sample.

1a. While the two constructs are highly correlated, race is socially defined while genetic-ancestry is biologically defined; and conflating the two does harm to both scientific knowledge and efforts to reduce the continuing harms caused by systemic racism (Phelan and Link 2013). As such, the following is strongly worded due to our history as both a species and scientists who include both "genetic" and "social" measures in our research. Please note that I do not think the author(s) intend their description of the very technical and important aspects of genomic analyses to reify race as a genetically defined construct, but as currently written, that's what some of their writing does.

For example, in the "Samples" section of the manuscript, the authors state that "we applied the same inclusion criteria to HSE data as used for the UKBB recruitment, retaining only individuals aged between 40 and 69 years and of (self-reported) European ancestry." ... Unless these self-reports are based on assessments of genetic ancestry conducted by a third party (e.g., 23andMe), this language must be changed to accurately reflect what is being measured (i.e., self-reported race/ethnicity).

A second example of the authors failure to make this distinction occurs in the section titled "Probability weighted genome-wide association analyses" where they write, "Additional QC filters for genome-wide analyses were applied to select participants (i.e., exclusion of related individuals, exclusion of non-White British ancestry based on principal components, ..." (emphasis added).

Note to the author(s): Those of us who pursue the expansion of knowledge in this area of science must be more careful in our use of language generally. While none of us write perfectly, we all must avoid applying terms used to, and/or created for the purpose of defining racial categories (categories which in-turn were/are used for the suppression and harm, legal and otherwise, of individuals while describing the current geo-political region(s) of the world that, on average, most closely

correlate with a person's own genetic variation.

2. The authors need to be careful about over stating the representativeness of the analytic data and results. Specifically, data restricted to individuals of western European genetic ancestry who also self-identify as White are not representative of the UK population in the 21st century. Instead this restriction represents a subset of the population and the language used to describe it should reflect that fact.

2a. For example, despite the fact that summary statistics from the UK Census data match those in the HSE data for the auxiliary variables, both data sources are restricted to a non-representative population of the UK (i.e., what the authors refer to as "White British ancestry"). As such, describing the sample and analysis as representative of the UK is misleading at best and purposefully obfuscating at worst.

2b. Similarly, I did not see any statement of if sampling weights were used in the comparison of the UK Census micro-data and HSE data (if I missed this please accept my apologies). This is particularly important, when the restriction criteria are added to the HSE (resulting in a 72% reduction in the HSE sample size).

3. In the "Samples" section of the manuscript, the authors need to do more to discuss why they use list-wise deletion instead of multiple imputation (or any other missing data corrections) to address missingness on the auxiliary variables.

4. I appreciate the authors attempt to make comparisons to some of the most highly powered GWAS phenotypes. However, I'm concerned as to the comparability of their chosen measures with the phenotypes employed in published GWASs. For example, most, if not all, GWASs of educational attainment measure education in years of school completed, not the age at which formal education was concluded. As such the authors need to do more to either defend their choice of phenotype measures and/or match the measurement used in those GWASs.

5. More needs to be said about the different clumping criteria used in GWAS ($kb = 250$ and an r^2 of 0.1) vs. Mendelian Randomization ($kb = 10,000$ and an r^2 of 0.001). This could be as simple as referencing standard procedures for each method or a more detailed discussion.

6. I am very confused about what the $N_{\text{effective}}$ is for the GWA results. For example, it appears that the N_s are negative as reported in the "Probability weighted genome-wide association analyses on UK Biobank traits" section of the "Results" section (e.g., the authors state, "[wGWA] $N_{\text{effective}} = 94,643-102,215$)... However, the $N_{\text{effective}}$ reported for the wGWA in the next section is listed as 102,215... Please revise with correct, and consistent, numbers.

Minor comments:

1. The authors should add a citation defending their decision to use a linear probability model instead of logistic or probit regression methods (e.g., Paul von Hippel's comments on Paul Allison's statistical consulting website Statistical Horizons: (1) <https://statisticalhorizons.com/when-can-you-fit/> , (2) <https://statisticalhorizons.com/linear-vs-logistic/>)

2. In the "Samples" subsection of the "Results" section of the manuscript the authors use percentages for some, but not all, of the #s to describe mean age when individuals completed full time education. Given the measure, I assume the values represent mean ages and not percentages.

Reviewer #3:

Remarks to the Author:

1. This is a very interesting and useful paper. My main concern is that the tone of the paper implies that the weighted analysis is necessarily less biased than the unweighted analysis. The authors acknowledge in the discussion that this is not the case, but I feel this is a crucial point that affects the whole paper. Weighting only removes bias if the weighting model is correct, with respect to the variables related to those in the analysis model. Thus, for each analysis, the implications of a particular

mis-specification in the weighting model are different. The mis-specification could arise due to variables being missed out of the weighting model, or mis-specification of the parametric form of the weighting model.

2. Variables being missed out is a plausible limitation here – the variables to be used were chosen for convenience, i.e. those that were available and could be harmonized across both datasets. This limitation is briefly mentioned in the discussion, but its implications (e.g. plausible size of bias?) are not mentioned.

3. Similarly, mis-specification of the weighting model could be an issue – e.g. although all two-way interactions are examined, I think the continuous variables are entered into the model as linear terms only.

4. I don't necessarily think the authors should make their weighting models any more complex than they already are, I just think that the limitations need to be more clearly acknowledged throughout the paper. For example, throughout the abstract the results are described as though the unweighted results are necessarily biased and the weighted are unbiased.

5. I would prefer to see the paper presented more as a sensitivity analysis for selection bias, rather than an absolute removal of the bias. I.e. the "truth" of the weighted results still depends on the assumption that the weighting model is correct, and this cannot be verified given the observed data.

6. As an extreme example, if the analysis model is a regression of y on x , where both x and y affect selection into UKBB, but x and y are unrelated to the 14 harmonised variables, then I think using the weights as derived would not affect the estimated coefficient for y on x – and yet it would be biased due to selection. You couldn't tell this from any of the observed data, I don't think – because you can only compare balance etc for the 14 harmonised variables.

7. The sex-participation analyses are very useful and do help somewhat to address point 6, as they are additional information.

8. I don't really see the added benefit of the GWA of the participation probabilities, over and above a GWA of the individual variables making up the predicted probabilities?

9. This claim, for example, I think is over-stated: "In summary, probability weighting successfully created a pseudo-sample of the UKBB achieving higher levels of representativeness, thus providing a useful tool for participation bias correction in downstream analyses.". The pseudo-sample is more representative (of the UK population presumably?) but only in the variables you can observe and compare. You can't know this is a useful tool to correct bias in all possible analyses.

10. It needs to be clear that figure 1 is not an exhaustive representation of all the ways that participation bias can arise in a genetic study.

Author Rebuttal to Initial comments

Reviewer 1

R1.1.

It would be valuable to mention that, yes, epidemiological studies are typically subjected to healthy-volunteer bias, but that cannot be said for many hospital-based biobanks which are common, especially in US (but also FinnGen). The biobanks are more likely to enroll sicker individuals.

We thank the reviewer for highlighting that selection mechanisms are likely to differ across biobanks. We have now extended the discussion to illustrate this more concretely for hospital-based biobanks:

Discussion (Manuscript): “Bias due to selective participation will differ across study contexts, and participation mechanisms as evaluated in this study are therefore not generalizable to other cohorts. For example, large health registry-based biobanks, where older individuals with poorer health tend to be over-represented, do not show the ‘healthy-volunteer bias’ (that characterises the UKBB), but those also show selection bias²⁷.”

Reference:

27. Lee J, Jukarainen S, Dixon P, et al. Quantifying the causal impact of biological risk factors on healthcare costs. *medRxiv*. Published online January 1, 2022:2022.11.19.22282356. doi:10.1101/2022.11.19.22282356

R1.2.

It would be valuable to clarify which variables are self-reported vs objectively measured. For example in HSE was BMI self-reported and was this compared to measured BMI in UK Biobank?

This is an important point. The only variables not based on self-reports were height/weight (to derive BMI) and level of urbanisation. These variables were measured in the same way in the UKBB and the HSE: In the UKBB, weight and height measures were taken during the initial assessment centre visit. In the HSE, height and weight measurements were taken during the face-to-face household interviews. Urbanisation in both the UKBB and HSE was derived by combining each participant’s home postcode with data generated from the 2001 Census from the Office of National Statistics. To make this clearer to the reader, we have added this information in the Supplement Table listing all UKBB/HSE measures used in this study. The table now also indicates which variables are based on self-report measures, and which are objectively measured.

Coding of variables included from the UKBB and HSE

Variable	UKBB	HSE	Coding	H	G
Frequency of alcohol use (SR)	About how often do you drink alcohol? (ID: 1558)	How often have you had an alcoholic drink of any kind during the last 12 months?	0=never, 1=few times/year, 2=few times/year, 3=monthly, 4=once or twice/week, 5=three or four days/week, daily	X	
Weekly alcohol use (SR)	In an average week, how many beer/cider/champagne/ wine/spirits/other alcohol) would you drink? (ID: 1588, 1578, 1608, 5364, 1568, 1598)		continuous		X
Physical activity (SR)	Number of days/week of vigorous physical activity 10+ minutes (ID:904)		continuous		X
Sex	Sex of participant	Sex of participant	0=Male/1=Female	X	X
Age	Age of participant	Age of participant	Continuous	X	
Years of education (SR)	At what age did you complete your continuous full-time education? (ID: 845) [note: Individuals with a University degree (ID: 6138) were allocated '19 or over']	At what age did you finish your continuous full-time education at school or college?	14 (or under) /15/16/17/18/19 or over	X	X
Smoking status (SR)	Summary if the current/past smoking status of the participant (ID: 20116)	Have you ever smoked a cigarette, a cigar or a pipe? / Do you smoke nowadays?	0=Never/1=previous/2=current	X	X

Vegetable intake (SR)	About how many heaped tablespoons of cooked vegetables would you eat per day? (ID: 1289)		Continuous		X
Fruit intake (SR)	About how many pieces of fresh fruit would you eat per day? (ID: 1309)		Continuous		X
Income (SR)	What is the average total income before tax received by your household? (ID: 738)	What is your household's income before any deductions for income tax, National Insurance, etc?	1=18k, 2=18k-31k, 3=31k-52k, 4=52k-100k, 5=>100k	X	
Household size (SR)	Including yourself, how many people are living together in your household? (ID: 705)	Interviewer collects the names of the people in the household	Categorical, with each category indicating the number of individuals living in the household. 7 = 7 individuals or more.	X	
Employment status (SR)	Which of the following describes your current situation? [...] (ID: 6142)	Which of these descriptions applies to what you were doing last week? [...]	1=unemployed, 2=employed, 3=economically inactive, 4=retired	X	
Height (M)	Measured using a Seca 202 device (ID: 50)	Measured during the face-to-face household interview	Continuous	X	X
Weight (M)	Measured during the initial Assessment Centre visit (ID: 210002)	Measured during the face-to-face household interview	Continuous	X	
BMI (M)	Derived from height and weight measures	Derived from height and weight measures	Continuous	X	X
BMI (categorical) (M)	Derived from height and weight measures	Derived from height and weight measures	1=underweight (BMI<18.5), 2=normal weight (18.5 < BMI <25) 3=overweight (25 < BMI <30) 4=obese (BMI ≥ 30),	X	
Non-cancer diagnoses (number) (SR)	Number of self-reported non-cancer illnesses (ID: 135)		Continuous		X
Risk taking (SR)	Would you describe yourself as someone who takes risks? (ID: 2040)		0=No/1=Yes		X
Loneliness (SR)	Do you often feel lonely? (ID: 2020)		0=No/1=Yes		X
Diabetes (SR)	Has a doctor ever told you that you have diabetes? (ID: 2443)		0=No/1=Yes		X
Depression/Anxiety (SR)	Seen a psychiatrist for nerves, anxiety, tension or depression (ID: 2100)		0=No/1=Yes		X
LDL (M)	LDL cholesterol (ID: 30780)		Continuous		X
SBP (M)	Systolic blood pressure, automated reading (ID: 4080)		Continuous		X
Reaction time (T)	Reaction time (mean time to correctly identify matches) (ID: 20023)		Continuous		X
Insomnia (SR)	Do you have trouble falling asleep at night or do you wake up in the middle of the night? (ID: 1200)		1=Never/rarely, 2=sometimes, 3=usually		X
Cancer (SR)	Has a doctor ever told you that you have had cancer? (ID: 2453)		0=No/1=Yes		X
Coffee intake (SR)	How many cups of coffee do you drink each day? (ID: 1498)		Continuous		X
Urbanisation (D)	Classification derived by combining each participant's home postcode with data generated from the 2001 census from the Office of National Statistics (ID: 20118)	Degree of urbanisation	1=village/hamlet, 2=town/fringe, 3=urban	X	
Overall health (SR)	In general, how would you rate your overall health? (2178)	How is your health in general?	1=poor; 2=fair, 3=good	X	

D=derived using postcode data; H=used for harmonization and included in the model predicting participation probability; G=included as outcome in genome-wide analyses; M=measured using objective devices; SR=self-reported; T=tested using computerized cognitive test.

R1.3.

Because you use LDAK software, wouldn't make more sense to calculate h^2 based on LDAK heritability model assumptions? Or at least using stratified LD-score to get more accurate estimates?

We used LDAK to perform classical linear regression only. As such, there are no specific model assumptions beyond those that apply to any genome-wide association study using linear regression analysis (e.g., as performed in PLINK, BOLT-LMM or REGENIE).

R1.4.

Figure 1, panel A: A little bit misleading that the distribution is truncated, even if mentioned in the legend. Can you add a symbol to clearly indicate truncation?

We thank the reviewer for this suggestion – the truncation is now indicated by an asterisk in the density curve:

R1.5.

Can the authors comment more extensively about the smaller effective sample size due to inverse probability weighting? Is this reduction a function of the sample size of the representative study used to derive weights? Or what determines the reduction in effective sample size? This is a rather important consideration if weighted GWAS should be implemented.

Inverse probability weighting will always lead to a reduction in the effective sample size (n_{eff}), provided there is variability among sampling weights. More specifically, the degree of sample size reduction is a function of the variability among sampling weights – the larger the spread, the larger the reduction in n_{eff} . Consider a sample of $n=2$ where individual one has a normalized weight of $w_{n1}=0.5$ (i.e., is < 1 , so will be down-weighted) and individual two has a normalized weight of $w_{n2}=1.5$ (i.e., is > 1 , so will be

up-weighted). Applying the formula used for effective sample size estimation [$\sum(w_n)^2/\sum(w^2)$] gives us an effective sample size of $n_{\text{eff}}=1.6$, highlighting that weighting led to a 20% reduction in sample size. Sampling weights with a lower spread would result in a larger effective sample size [e.g., $w_{n1}=1.1$; $w_{n2}=0.9$; $n_{\text{eff}}=1.98$] and sampling weights with no variability will recover the original sample size of $n=2$ [e.g., $w_{n1}=1$; $w_{n2}=1$; $n_{\text{eff}}=2$].

As such, when deriving sampling weights, there is always a bias-variance tradeoff, as weights that maximize n_{eff} (e.g., by applying trimming to reduce the variability of the weights) will less efficiently correct for bias. Since our aim was to maximize bias correction, we abstained from applying correction methods to maximize n_{eff} , resulting in relatively large variabilities of the sampling weights (cf., legend Figure 1: "Panel (A) presents the truncated density curves of the normalized probability weights (w_{in}) for UKBB participants, ranging from 0.02 to 50.01.") and, hence, substantially reduced n_{eff} .

We have now added a brief mention of the key determinants of the effective sample:

Results (Manuscript): "The effect of participation bias on genome-wide results was evaluated by comparing the SNP estimates obtained from probability weighted genome-wide analyses, wGWA ($\hat{\beta}_w$, $N_{\text{effective}} = 94,643$ to $102,215$), to standard GWA analyses ($\hat{\beta}$, $N=263,464$ to $283,749$). Note that reductions in the effective sample size result from variability among the probability weights: when the weights are normalised to have a mean of one, the effective sample size simplifies to $n^*(1/(\text{Var}(w_{in})+1))$. Thus, this quantity depends on the unweighted study sample size (n) and on the variance of the normalised weights (w_{in})."

R1.6.

Related to the comment above. Are the 1.48% new SNPs from wGWAS identified despite the reduction in effective sample size?

This is indeed the case – the identified SNPs were picked up despite the reductions in effective sample size, but thanks to the increased estimated effect size.

R1.7.

The Manhattan plot for the GWAS of the liability to UKBB participation doesn't look very robust. Did you filter out rare variants? For example, rs189384883 does not seem to have LD partners with similar statistical significance. Many of the signals do not have the traditional "GWAS peak". I would be careful about interpreting individual SNPs obtained from this analysis.

It is true that most of the 23 clumped SNPs associated with UKBB participation were not in LD with nearby SNPs (only 5 SNPs genome-wide significant hits were removed via clumping).

While we excluded rare variants based on $\text{MAF} < 1\%$, all but one (rs16977020) of these 23 SNPs were low-frequency variants ($1\% < \text{MAF} < 5\%$). Thus, we agree that a careful interpretation of these

variants is warranted. Since the primary purpose of this GWA was to assess the genetic correlations between the UKBB participation with other traits (including participatory behaviours), rather than the interpretation of individuals variants associated with UKBB participation, we have now removed the results from the main text. Since some readers may nevertheless be interested in looking up additional information of the SNPs associated with UKBB participation, we have now included the results in the Supplement.

Results (Manuscript): “28 SNPs reached genome-wide significance ($p < 5 \times 10^{-8}$), of which 23 LD-independent SNPs were selected after clumping. Supplementary figures (Manhattan and QQ plot) and information (gene and phenotype annotation) for these SNPs is available in sFigure5-6 and sTable 5-6. ~~Figure 4A shows the Manhattan plot with positional mapping of genome-wide SNPs associated with the liability to UKBB participation (cf. sTable 6 for annotation and estimates of significant SNPs). The QQ plot (sFigure 5) can be found in the Supplement. SNP heritability for UKBB participation was $h^2=0.009$ (se=0.005) (LD-score intercept: 1.055). A lookup of SNP-trait associations estimated in previous GWA analyses showed that UKBB participation associated variants mostly tapped into age-related outcomes (e.g., cause of death: cancer/dementia/fatty liver disease/pneumonia) (sTable 7).~~

Reviewer 2

Below is my review of manuscript NATHUMBHAV-22092529 titled, “Correction for participation bias in the UK Biobank reveals non-negligible impact on genetic associations and downstream analyses.” This study seeks to investigate if the use of inverse-probability weights from a representative sample (Health Survey England) applied to a non-representative volunteer-based sample (the UK Biobank) can improve estimates obtained from three types of genomic analyses (GWAS, LD score regression, Mendelian Randomization). Importantly, this study addresses ongoing weakness in both genetic discovery and the application of those discoveries via PGSs and other methods in population-level research.

We thank the reviewer for this summary of our study. R2.1.

The author(s) need to better clarify the distinction between race/ethnicity and genetic ancestry and when/how they use both to define their analytic sample.

While the two constructs are highly correlated, race is socially defined while genetic-ancestry is biologically defined; and conflating the two does harm to both scientific knowledge and efforts to reduce the continuing harms caused by systemic racism (Phelan and Link 2013). As such, the following is strongly worded due to our history as both a species and scientists who include both “genetic” and “social” measures in our research. Please note that I do not think the author(s) intend their description of the very technical and important aspects of genomic analyses to reify race as a genetically defined construct, but as currently written, that’s what some of their writing does.

For example, in the “Samples” section of the manuscript, the authors state that “we applied the same inclusion criteria to HSE data as used for the UKBB recruitment, retaining only individuals aged between 40 and 69 years and of (self-reported) European ancestry.” ... Unless these self-reports are based on assessments of genetic ancestry conducted by a third party (e.g., 23andMe), this language must be changed to accurately reflect what is being measured (i.e., self-reported race/ethnicity).

A second example of the authors failure to make this distinction occurs in the section titled “Probability weighted genome-wide association analyses” where they write, “Additional QC filters for genome-wide analyses were applied to select participants (i.e., exclusion of related individuals, exclusion of non-White British ancestry based on principal components, ...” (emphasis added).

Note to the author(s): Those of us who pursue the expansion of knowledge in this area of science must be more careful in our use of language generally. While none of us write perfectly, we all must avoid applying terms used to, and/or created for the purpose of defining racial categories (categories which in-turn were/are used for the suppression and harm, legal and otherwise, of individuals while describing the current geo-political region(s) of the world that, on average, most closely correlate with a person’s own genetic variation.

We could not agree more with the reviewer’s view concerning the careful use of terms relating to race/ethnicity and genetic ancestry, and we apologize for not having included a more nuanced distinction between these terms in the initial manuscript. We therefore thank the reviewer for giving us the opportunity to provide clearer and more consistent definitions of these concepts. We have now revised the sections accordingly:

Methods (Manuscript): “Included in this study were data from UK Biobank participants of **European genetic ancestry who also self-identify as White** and passed standard GWA analysis quality control measures [...].”

Methods (Manuscript): “We applied the same inclusion criteria to HSE data as used for UKBB recruitment, retaining only individuals aged between 40 and 69 years **who self- identify as White** [...].”

Methods (Manuscript): “Additional QC filters for genome-wide analyses were applied to select participants (i.e., restricting the sample to unrelated individuals of **European genetic ancestry** [...].”

Methods (Manuscript): “We applied the same selection criteria to the Census data as to the UKBB and HSE (i.e., filtered according to geographical region, **ethnic group** and age) [...].”

Results (Manuscript): “[...] we excluded individuals of age >69 and <40 (n=2463), individuals from Scotland or Wales (n=56,483), individuals who **self-identify as non- White** (n=28,371) [...].”

Discussion (Manuscript): “[...] the genome-wide results discussed here can only be generalized to

adults of European genetic ancestry who also self-identify as White.”

R2.2.

The authors need to be careful about over stating the representativeness of the analytic data and results. Specifically, data restricted to individuals of western European genetic ancestry who also self-identify as White are not representative of the UK population in the 21st century. Instead this restriction represents a subset of the population and the language used to describe it should reflect that fact.

For example, despite the fact that summary statistics from the UK Census data match those in the HSE data for the auxiliary variables, both data sources are restricted to a non- representative population of the UK (i.e., what the authors refer to as “White British ancestry”). As such, describing the sample and analysis as representative of the UK is misleading at best and purposefully obfuscating at worst.

We fully agree that the subsamples selected from the UKBB, HSE and Census data are by no means representative of the UK population as a whole. We apologize if we gave the impression that we aimed for representativeness of the whole UK population, as this was not possible given the data at hand (e.g., exclusion of individuals younger than 40 or older than 70 in the UKBB). What we tried to convey instead is that we aimed to make the UKBB more representative of its target population – a defined population where each individual has the same probability of being sampled. The UKBB target population is very different from the UK population as a whole (e.g., targeting only middle-aged and older adults in the UK, rather than all age groups living in the UK). The major goal of our study was to show how genetic association study results (and the downstream analyses using those associations) can be biased due to study participation. Hence, by default, we aimed to derive less biased estimates that we would have obtained, had we sampled all UK individuals of European genetic ancestry who also self-identify as White aged between 40 and 69 years. This is our target population, from which HSE was sampled (with reasonably high participation rate).

Genetic association studies conducted in mixed ancestry groups lead to much more biased estimates than what participation bias would lead to, hence the pre-selected ancestry group was very important. We have tried to emphasize in the initial draft of manuscript that representativeness refers to a defined target population (rather than the total UK population) – a few examples are below:

Abstract: “While large-scale volunteer-based studies such as the UK Biobank (UKBB) have become the cornerstone of genetic epidemiology, the study participants are rarely representative of their target population.”

Introduction (Manuscript): “A particularly challenging bias – and typically not considered in genetic studies – can occur when data is collected from individuals not representative of their target

population”

Introduction (Manuscript): “participants typically show a better health profile than the target population”

Introduction (Manuscript): “Participation bias is eliminated by the use of samples that are representative of their target population. To increase representativeness in the UKBB, we derive a model for participation probability and create a pseudo-sample of the UKBB matching its target population”

Method (Manuscript): “This approach [probability weighting] adjusts for non-response bias by weighting over- and under-represented individuals, thereby creating a pseudo- population that is more representative of its target population”

Method (Discussion): “Biobank data in which participation bias cannot be assessed (e.g., in self-selection samples without a defined target population) may therefore be only of limited utility when scrutinizing genotype-phenotype relationships.”

However, to avoid any misunderstanding of the notions concerning representativeness and target population, we have now provided additional information in the Supplement:

Introduction (Manuscript): “Participation bias is eliminated by the use of samples that are representative of their target population (cf. Supplement note for a definition of target population).”

Supplement:

“A note on target population and representativeness

A target population is a broader group of individuals from which a study sample is drawn and to which the study results should generalize to. Depending on the research question, this might be babies born in India in 2007¹, US woman diagnosed with breast cancer², or – as is the case for the UK Biobank – middle-aged to older adults living in the United Kingdom. A representative sample is a subset of that target population that accurately reflects the properties of that group. To ensure representativeness of the group of individuals sampled from the target population, each individual must have the same chance of being included in the sample.”

References:

1. Singh A, Yadav A, Singh A. Utilization of postnatal care for newborns and its association with neonatal mortality in India: An analytical appraisal. *BMC Pregnancy Childbirth*. 2012;12(1):33. doi:10.1186/1471-2393-12-33
2. Darby SC, McGale P, Taylor CW, Peto R. Long-term mortality from heart disease and lung cancer after radiotherapy for early breast cancer: prospective cohort study of about 300 000 women in US SEER cancer registries. *Lancet Oncol*. 2005;6(8):557-

565. doi:[https://doi.org/10.1016/S1470-2045\(05\)70251-5](https://doi.org/10.1016/S1470-2045(05)70251-5)

Finally, in response to the reviewer's comment ("this restriction [to individuals of European genetic ancestry] represents a subset of the population and the language used to describe it should reflect that fact."), we have now highlighted more explicitly that the restriction to middle-aged and older adults of European genetic ancestry who also self-identify as White is a limitation of most studies (including ours) making use of the UKBB for genomic research.

The discussion now reads as follows:

Discussion (Manuscript): "Bias due to selective participation will differ across study contexts, and participation mechanisms as evaluated in this study are therefore not generalizable to other cohorts. [...] the genome-wide results discussed here can only be generalized to adults of European genetic ancestry who also self-identify as White.

Future work should also assess the impact of participation bias in more diverse samples, notably other ancestries and ethnic groups, as well as younger individuals, once large-enough samples become available."

Similarly, I did not see any statement of if sampling weights were used in the comparison of the UK Census micro-data and HSE data (if I missed this please accept my apologies). This is particularly important, when the restriction criteria are added to the HSE (resulting in a 72% reduction in the HSE sample size).

The reduction in HSE sample size results from selecting a subset of the HSE data to match the target population criteria defined by the UKBB (e.g., individuals aged between 40 and 69 years). Since all individuals included in the HSE subset still have the same inclusion probabilities, no probability weights need to be used for this subset of individuals. This is to highlight that selecting a target sample is different to reduction in sample size as a result of, for example, attrition. Here, mechanisms for drop-out will vary across individuals, resulting in probability weights that differ across selected individuals.

R2.3.

In the "Samples" section of the manuscript, the authors need to do more to discuss why they use list-wise deletion instead of multiple imputation (or any other missing data corrections) to address missingness on the auxiliary variables.

We thank the reviewer for raising this important point. We believe that missing data in auxiliary variables is unlikely to have caused problems in our study, for two main reasons. First, to facilitate complete-case analysis, we restricted the selection of auxiliary variables to those with only few missing data points. As shown in the table below, missing data in auxiliary variables in the UKBB ranged from 0% (age, sex) to 2.1% (education). In the HSE, missing data ranged from 0% to 2% (BMI). As a results, we

removed 21,868 UKBB participants (5.27% of the target sample) and 830 HSE participants (3.37% of the target sample) with missing data in any of the auxiliary variables. As the proportion of missing data is low, we believe that list-wise deletion is unlikely to have resulted in bias. Second, inverse probability weighting (IPW) is a method designed to remove bias due to missing data. Since the 21,868 UKBB participants with missing data can be considered as a special case of ‘missingness due to non-participation’, the application of IPW would compensate for this fraction of missing data, in particular since our generated probability weights showed high genetic correlations (r_g) with other participatory behaviour (e.g., $r_g=0.75$ for GWA on ‘provided e-mail address for re-contact’ and our GWA on UKBB participation). Hence, our approach is logical since it treats samples with missing data as if they had not participated and derives the weights in accordance. Since genetic associations cannot be computed for individuals with missing phenotypes, this is a fair use of the data. Indeed, imputation could have been used, but those imputed values would have considerably larger variance hence would have to be treated differently. On the other hand, multiple imputation for the GWA would have been computationally too expensive to consider.

As proposed by the reviewer, we have now extended the ‘Samples’ section in the manuscript to make this clearer to the reader.

Results/Samples (Manuscript): “Further removed were 21,868 (5.27%) individuals with missing data in any of the auxiliary variables. Since these individuals can be considered a special case of missingness due to non-participation, which the probability weights were designed to compensate for, we did not impute missing data for auxiliary variables.”

Auxiliary variable	n (%) missing in UKBB	n (%) missing in HSE
Age	0 (0%)	0 (0%)
Alcohol frequency	296 (0.07%)	7 (0.03%)
BMI	1875 (0.45%)	447 (1.97%)
Education (age)	8662 (2.09%)	70 (0.31%)
Employment status	3412 (0.82%)	16 (0.07%)
Height	1429 (0.34%)	120 (0.53%)
Household size	2480 (0.6%)	0 (0%)
Income	4643 (1.12%)	343 (1.51%)
Overall health	1634 (0.39%)	9 (0.04%)
Sex	0 (0%)	0 (0%)
Smoking status	1502 (0.36%)	2 (0.01%)
Urbanisation	3783 (0.91%)	0 (0%)

Weight	1716 (0.41%)	430 (1.9%)
--------	--------------	------------

R2.4.

I appreciate the authors attempt to make comparisons to some of the most highly powered GWAS phenotypes. However, I'm concerned as to the comparability of their chosen measures with the phenotypes employed in published GWASs. For example, most, if not all, GWASs of educational attainment measure education in years of school completed, not the age at which formal education was concluded. As such the authors need to do more to either defend their choice of phenotype measures and/or match the measurement used in those GWASs.

It is true that GWA studies using UKBB data have used slightly different measures to index educational attainment, mostly relying on self-reported 'educational qualification' to derive years of schooling [Okbay et al. (2016¹/2022²)] or to derive binary outcome measures indexing higher education (yes/no) [e.g., Rietveld et al. (2013)³; Rask-Andersen et al. (2021)⁴].

For the purpose of our study, we have chosen a measure that could be directly harmonized and compared with the HSE data. The measure 'age when completed continuous full-time education' was particularly useful in this context, as it requires fewer 'best guesses' about the number of years of schooling that correspond to a specific educational qualification. For example, inferring the number of years of schooling for UKBB individuals stating 'None of the above' or HSE individuals reporting 'foreign/other' as their highest educational qualification likely induces measurement error. Importantly, we would think that our phenotype derived from 'age when completed full-time education' is very similar to that used in the most recent GWA on educational attainment, as their phenotype was partly derived using that same variable (e.g., as noted in the study (cf. Ref²): "UKB participants with a qualification of 'NVQ or HND or HNC or equivalent' [...] were previously coded as having 19 years of education, but this classification overstates their average years of schooling [...]. We therefore recoded EduYears for these participants as the age they reported leaving full-time education minus five."). Nevertheless, we conducted LD score regression analysis to evaluate the convergence between our GWA and the most recent GWA study on educational attainment, highlighting that both GWA are highly correlated ($r_g=0.96$) (cf. table below).

With respect to the other phenotypes included in our study, we believe that there is very little heterogeneity in the outcome definitions between our study and previously published genome-wide studies. First, we included a number of objectively measured phenotypes (BMI, height, weight, LDL cholesterol, systolic blood pressure), which are directly comparable across studies. Second, results from LD score regression (cf. table below) further highlight that the convergence between our GWA results and previously published GWA studies on self-reported outcomes is high. Genetic correlation estimates for selected phenotypes are large (all $r_g>0.9$), even for complex behavioural phenotypes such as alcohol use ($r_g=0.98$).

In summary, we believe that our GWA outcomes are highly comparable to those previously published using UKBB and other genomic datasets.

Phenotype (published study)	Phenotype (present study)	r_g
Major depressive disorder (n=173,005) Wray et al. (2018) ⁵	Depression/Anxiety	0.96
Insomnia (n= 113,006) Hammerschlag et al. (2017) ⁶	Insomnia	0.96
BMI (n=681,275) Yengo (2018) ⁷	BMI	0.97
Educational attainment (n=765,283) Okbay et al. (2022) ²	Education	0.96
Height (n=693,529) Yengo (2018) ⁷	Height	0.97
Loneliness (n=487,647) Day (2018) ⁸	Loneliness	1.02
Alcohol use (drinks per week) (n=513,208) Liu (2019) ⁹	Alcohol use (drinks per week)	0.98

References

- 1 Okbay A, Beauchamp JP, Fontana MA, Lee JJ, Pers TH, Rietveld CA *et al.* Genome-wide association study identifies 74 loci associated with educational attainment. *Nature* 2016; 533: 539–542.
- 2 Okbay A, Wu Y, Wang N, Jayashankar H, Bennett M, Nehzati SM *et al.* Polygenic prediction of educational attainment within and between families from genome-wide association analyses in 3 million individuals. *Nat Genet* 2022; 54: 437–449.
- 3 Rietveld CA, Medland SE, Derringer J, Yang J, Esko T, Martin NW *et al.* GWAS of 126,559 individuals identifies genetic variants associated with educational attainment. *Science (80-)* 2013; 340: 1467–1471.
- 4 Rask-Andersen M, Karlsson T, Ek WE, Johansson Å. Modification of heritability for educational attainment and fluid intelligence by socioeconomic deprivation in the UK Biobank. *Am J Psychiatry* 2021; 178: 625–634.
- 5 Wray NR, Ripke S, Mattheisen M, Trzaskowski M, Byrne EM, Abdellaoui A *et al.* Genome-wide association analyses identify 44 risk variants and refine the genetic architecture of major depression. *Nat Genet* 2018; 50: 668–681.
- 6 Hammerschlag AR, Stringer S, de Leeuw CA, Sniekers S, Taskesen E, Watanabe K *et al.* Genome-wide association analysis of insomnia complaints identifies risk genes and genetic overlap with psychiatric

- and metabolic traits. *Nat Genet* 2017; 49: 1584–1592.
- 7 Yengo L, Sidorenko J, Kemper KE, Zheng Z, Wood AR, Weedon MN *et al.* Meta-analysis of genome-wide association studies for height and body mass index in ~700000 individuals of European ancestry. *Hum Mol Genet* 2018; 27: 3641–3649.
 - 8 Day FR, Ong KK, Perry JRB. Elucidating the genetic basis of social interaction and isolation. *Nat Commun* 2018; 9: 2457.
 - 9 Liu M, Jiang Y, Wedow R, Li Y, Brazel DM, Chen F *et al.* Association studies of up to 1.2 million individuals yield new insights into the genetic etiology of tobacco and alcohol use. *Nat Genet* 2019; 51: 237–244.

R2.5.

More needs to be said about the different clumping criteria used in GWAS (kb = 250 and an r^2 of 0.1) vs. Mendelian Randomization (kb = 10,000 and an r^2 of 0.001). This could be as simple as referencing standard procedures for each method or a more detailed discussion.

We thank the reviewer for this suggestion – we have now included references for the clumping parameters in the main text:

Methods (Manuscript): “LD-independent SNPs reaching genome-wide significance ($p < 5 \times 10^{-8}$) were selected via clumping (--clump-kb 250 --clump-r2 0.1, following standard recommendations⁴⁰).”

40. Adam Y, Samtal C, Brandenburg J, Falola O, Adebisi E. Performing post-genome-wide association study analysis: overview, challenges and recommendations.

F1000Research. 2021;10:1002. doi:10.12688/f1000research.53962.1

“As genetic instruments, we selected LD-independent (--clump-kb 10,000 --clump-r2 0.001, adhering to standard MR protocols⁴⁷) SNPs reaching genome-wide significance ($p < 5 \times 10^{-8}$) [...]”

47. Rasooly D, Patel CJ. Conducting a reproducible Mendelian Randomization analysis using the R Analytic Statistical Environment. *Curr Protoc Hum Genet*. 2019;101(1). doi:10.1002/cphg.82

R2.6.

I am very confused about what the $N_{\text{effective}}$ is for the GWA results. For example, it appears that the N s are negative as reported in the “Probability weighted genome-wide association analyses on UK Biobank traits” section of the “Results” section (e.g., the authors state, “[wGWA] $N_{\text{effective}} = 94,643 - 102,215$)... However, the $N_{\text{effective}}$ reported for the wGWA in the next section is listed as 102,215... Please revise with correct, and consistent, numbers.

We wish to clarify that the section quoted above does not include a negative estimate of the effective sample size. We assume that the reviewer has erroneously interpreted the dash in “ $N_{\text{effective}} = 94,643 - 102,215$)” as a minus sign. To avoid confusion, we have now replaced the dash sign with ‘to’ in the

relevant sections:

Abstract: “Comparing the output obtained from wGWA ($N_{\text{effective}}=94,643$ to 102,215) to standard GWA analyses [...]

Results (Manuscript): “The effect of participation bias on genome-wide results was evaluated by comparing the SNP estimates obtained from probability weighted genome-wide analyses, wGWA ($\hat{\beta}$, $N_{\text{effective}}=94,643$ to 102,215) [...]

R2.7.

Minor comment: The authors should add a citation defending their decision to use a linear probability model instead of logistic or probit regression methods (e.g., Paul von Hippel’s comments on Paul Allison’s statistical consulting website Statistical Horizons:

- (1) <https://statisticalhorizons.com/when-can-you-fit/>,
- (2) <https://statisticalhorizons.com/linear-vs-logistic/>)

We thank the reviewer for pointing us to the two references, which discuss situations where the linear model has certain advantages over the logistic model for a binary outcome. We have added the two references in the manuscript, in addition to a post by the Neale Lab (<http://www.nealelab.is/blog/2017/9/11/details-and-considerations-of-the-uk-biobank-gwas>), which specifically details the pros and cons of employing a linear model for binary outcomes when using UKBB data in genome-wide studies. The revised paragraph in the manuscript now reads as follows:

Methods (Manuscript): “We applied a linear model to all outcomes (continuous and binary traits). This was done to allow for standardization of SNP estimates and to ensure comparability of effect sizes. A more detailed discussion on the advantages and disadvantages of using a linear over a logistic model for binary outcomes is provided by von Hippel^{37,38} as well as the Neale Lab³⁹ discussing its application specifically when using UKBB data”

37. von Hippel P. Linear vs. Logistic Probability Models: Which is Better, and When? Published 2015. Accessed November 9, 2022. <https://statisticalhorizons.com/linear-vs-logistic/>
38. von Hippel P. When Can You Fit a Linear Probability Model? More Often Than You Think. Published 2017. Accessed November 9, 2022. <https://statisticalhorizons.com/when-can-you-fit/>
39. Howrigan D, Abbott L, Churchhouse C, Palmer D, Neale B. Details and considerations of the UK Biobank GWAS. Published 2017. Accessed November 9, 2022. <http://www.nealelab.is/blog/2017/9/11/details-and-considerations-of-the-uk-bioban>

k- gwas

R2.8.

Minor comment: In the “Samples” subsection of the “Results” section of the manuscript the authors use percentages for some, but not all, of the #s to describe mean age when individuals completed full time education. Given the measure, I assume the values represent mean ages and not percentages.

We had indeed erroneously added ‘%’ after two mean values – these are now removed:

Results (Manuscript): “mean age when completed full time education ($M_{\text{CENSUS}}=16.6$, $M_{\text{HSE}}=16.4$, $M_{\text{UKBB}}=17.2$)”

Reviewer 3

R3.1.

This is a very interesting and useful paper. My main concern is that the tone of the paper implies that the weighted analysis is necessarily less biased than the unweighted analysis. The authors acknowledge in the discussion that this is not the case, but I feel this is a crucial point that affects the whole paper. Weighting only removes bias if the weighting model is correct, with respect to the variables related to those in the analysis model. Thus, for each analysis, the implications of a particular mis-specification in the weighting model are different. The mis-specification could arise due to variables being missed out of the weighting model, or mis-specification of the parametric form of the weighting model.

We thank the reviewer very much for the positive and constructive feedback, which helped us to revise and extend the manuscript. We fully agree that model mis-specification resulting from variables being missed in the model, or the use of parametric weighting models that include incorrect functional forms of the auxiliary variables, would result in residual selection bias.

More specifically, with regard to the first point (variables being missed in the weighting model), we had indeed mentioned this in the discussion, but we recognize that more emphasize on this point would clarify some of the inherent limitations of participation bias correction methods.

Discussion (manuscript): “This study comes with a number of shortcomings. First, while the application of probability weighting successfully **removed reduced** bias resulting from selective participation in the UKBB, residual bias may still exist. Important factors independently predicting UKBB participation may have been missed when modelling participation probability, **as our auxiliary variables were limited to those measured in both the UKBB and the reference sample. For example, probability weighing would not correct bias in situations where the exposure and the outcome of interest both link to study**

participation, but are unrelated to the auxiliary variables. As such, the degree of participation bias documented in this study represents only the lower bound of the total bias induced by selective UKBB participation. Weighting – like any other method adjusting for non-representativeness – should therefore only be considered as the second-best option when tackling participation bias, as only probability sampling at the recruitment stage can ensure full elimination of this type of bias.”

With regard to the second point (‘mis-specification in the weighting model’), we believe that problems caused by non-linear effects in our study are negligible, considering our modelling approach. More specifically, whenever possible, we transformed continuous into categorical variables and included them as dummy variables alongside their continuous counterpart in the model. For example, the continuous variable ‘household size’ (ranging from 0 to the maximum number of individuals in a particular household) was categorized into 7 distinct classes, with 1 indexing a single-person household and 7 indexing a household of 7 or more. BMI was categorized into four distinct groups (underweight, normal weight, overweight, obese), according to international (World Health Organization) criteria. The remaining continuous variables included age, educational attainment, height and weight. As such, most of the variables included in the weighting models were not of continuous but categorical nature (10 out of 14).

However, we agree that non-linear effects of these four continuous variables could cause problems in the parametric approach employed in our study (LASSO regression). To evaluate further if misspecification of the functional term has led to inefficient sampling weights, we also tested the performance of weights derived from a nonparametric model (Random Forestⁱ). The table below contrasts the performance of sampling weight derived from LASSO regression and Random Forest, highlighting that LASSO regression outperformed Random Forest in terms of bias reductionⁱⁱ (97% versus 68% mean bias reduction, respectively) and efficiency ($n_{\text{Effective}}=141,249$ versus $n_{\text{Effective}}=105,600$, respectively). Given the high performance of LASSO regression, we believe that there is little evidence to suggest that the misspecification of the functional terms caused significant issues in the sampling weights.

	Bias reduction: Mean (range)	Effective sample size
LASSO regression	97% (range: 95% to 99.8%)	141,249
Random Forest	68% (range: -66% to 95%)	105,600

R3.2.

Variables being missed out is a plausible limitation here – the variables to be used were chosen for convenience, i.e. those that were available and could be harmonized across both datasets. This limitation is briefly mentioned in the discussion, but its implications (e.g. plausible size of bias?) are not mentioned.

While it is not possible to provide an estimate of the bias that has been missed by our correction method, we have now discussed in more detail the implications of residual bias in the revised version of the manuscript (cf. response to R3.1).

R3.3.

Similarly, mis-specification of the weighting model could be an issue – e.g. although all two- way interactions are examined, I think the continuous variables are entered into the model as linear terms only.

We thank the reviewer for raising this point. As already discussed in our response to R3.1, we found little evidence of issues relating to mis-specified of functional terms in LASSO regression.

R3.4.

ⁱ Package ‘RandomForest’ as implemented in R, using default parameters with n trees=500. Reference: Liaw, A., & Wiener, M. (2002). Classification and regression by randomForest. R news, 2(3), 18-22

ⁱⁱ Bias reduction was defined as described in the manuscript (Methods/Analysis): “We quantified participation bias as the differences between the correlations among all auxiliary variables within the UKBB (r_{UKBB}) and the HSE (r_{HSE}). The degree to which the weighted correlations (r_{UKBB_W}) reduced bias was estimated as $(|r_{HSE} - r_{UKBB}| - |r_{HSE} - r_{UKBB_W}|) / (|r_{HSE} - r_{UKBB}|)$, where a value of one indicates that weighting fully eliminated bias.”

I don't necessarily think the authors should make their weighting models any more complex than they already are, I just think that the limitations need to be more clearly acknowledged throughout the paper. For example, throughout the abstract the results are described as though the unweighted results are necessarily biased and the weighted are unbiased.

These are important points that we fully agree with. We had tried to convey in the manuscript that weighted GWA is a method to *reduce* bias rather than *fully eliminate* bias in most parts of the paper:

Abstract (Manuscript): "we propose a viable solution to reduce such bias"

Introduction (Manuscript): "Thereby, it is possible to evaluate how a shift towards representativeness impacts genome-wide findings and downstream analyses."

Results (Manuscript): "probability weighting successfully created a pseudo-sample of the UKBB achieving higher levels of representativeness"

However, we acknowledge that other parts of the manuscript were less explicit about the possibility that the application of bias correction methods may not fully eliminate participation bias. We have now amended the title, abstract and the main text as follows:

Title: ~~Correction for~~ The impact of participation bias in the UK Biobank on genetic associations and downstream analyses

Abstract (Manuscript): "Moving forward, increasing representativeness in biobank samples is paramount [...]"

Introduction (Manuscript): "To achieve increase representativeness in the UKBB, we derive a model for participation probability and create a pseudo-sample of the UKBB matching its target population."

Methods (Manuscript): "This approach adjusts for non-response bias by weighting over- and under-represented individuals, thereby creating a pseudo-population that is more representative of its target population."

Results (Manuscript): "Application of probability weighting ~~eliminated most~~ reduced bias induced by selective participation (median bias reduction: 0.97; mean: 0.91, range: 0.58 - 0.998)."

Results (Manuscript): "In summary, probability weighting successfully created a pseudo-sample of the UKBB achieving higher levels of representativeness, thus providing a useful tool for participation bias correction reduction in downstream analyses."

Discussion (Manuscript): “This study comes with a number of shortcomings. First, while the application of probability weighting successfully ~~removed~~ reduced bias resulting from selective participation in the UKBB, residual bias may still exist. Important factors independently predicting UKBB participation may have been missed when modelling participation probability, as our auxiliary variables were limited to those measured in both the UKBB and the reference sample. For example, probability weighing would not correct bias in situations where the exposure and the outcome of interest both link to study participation, but are unrelated to the auxiliary variables. As such, the degree of participation bias documented in this study represents only the lower bound of the total bias induced by selective UKBB participation. Weighting – like any other method adjusting for non-representativeness – should therefore only be considered as the second-best option when tackling participation bias, as only probability sampling at the recruitment stage can ensure full elimination of this type of bias.”

R3.5.

I would prefer to see the paper presented more as a sensitivity analysis for selection bias, rather than an absolute removal of the bias. I.e. the “truth” of the weighted results still depends on the assumption that the weighting model is correct, and this cannot be verified given the observed data.

We apologize if we gave the impression that the application of weighted GWA has led to an absolute removal of bias. We hope that our aim (reduction of bias) is now made more explicit in the revised version of the manuscript (cf. response to R3.4).

R3.6.

As an extreme example, if the analysis model is a regression of y on x , where both x and y affect selection into UKBB, but x and y are unrelated to the 14 harmonised variables, then I think using the weights as derived would not affect the estimated coefficient for y on x – and yet it would be biased due to selection. You couldn’t tell this from any of the observed data, I don’t think – because you can only compare balance etc for the 14 harmonised variables.

This point (residual bias due to missed variables in the weighting model) was already addressed in response to R3.1. In the revised discussion (cf., response to R3.1), we have now included the example provided by the reviewer in the comment above, to better illustrate why residual bias can persist after the application of probability weighting.

Discussion (Manuscript): “This study comes with a number of shortcomings. First, while the application of probability weighting successfully ~~removed~~ reduced bias resulting from selective participation in the UKBB, residual bias may still exist. Important factors independently predicting UKBB participation may have been missed when modelling participation probability, as our auxiliary variables were limited to those measured in both the UKBB and the reference sample. For example, probability weighing would not correct bias in situations where the exposure and the outcome of interest both link to study

participation, but are unrelated to the auxiliary variables. As such, the degree of participation bias documented in this study represents only the lower bound of the total bias induced by selective UKBB participation. Weighting – like any other method adjusting for non-representativeness – should therefore only be considered as the second-best option when tackling participation bias, as only probability sampling at the recruitment stage can ensure full elimination of this type of bias.”

R3.7.

The sex-participation analyses are very useful and do help somewhat to address point 6, as they are additional information.

We thank the reviewer for appreciating this sensitivity analysis. R3.8.

I don't really see the added benefit of the GWA of the participation probabilities, over and above a GWA of the individual variables making up the predicted probabilities?

Looking at the genetic correlations between UKBB participation and other traits was motivated by two main reasons. First, we were able to identify traits that most strongly correlate with UKBB participation, including those used to derive the UKBB participation probabilities. Interpreting the contribution of individual variables included in LASSO regression predicting UKBB participation is more challenging in this context, given the inclusion of interaction terms in the model. We therefore believe that a presentation in form of genetic correlations can provide a more comprehensive picture of correlates of UKBB participation. Second, we were able to look at genetic correlations with phenotypes not (readily) available in the UKBB, such as personality traits, participatory behaviour or mental health (e.g., schizophrenia, ADHD). While these results are not the main focus of the paper, we hope that they are nevertheless useful to the reader to gain insights into correlates of UKBB participation.

R3.9.

This claim, for example, I think is over-stated: “In summary, probability weighting successfully created a pseudo-sample of the UKBB achieving higher levels of representativeness, thus providing a useful tool for participation bias correction in downstream analyses.”. The pseudo-sample is more representative (of the UK population presumably?) but only in the variables you can observe and compare. You can't know this is a useful tool to correct bias in all possible analyses.

We have now tried to tone down the sentence quoted above as follows:

Results (Manuscript): “In summary, probability weighting successfully created a pseudo-sample of the UKBB achieving higher levels of representativeness, thus providing a useful tool for participation bias **correction** **reduction** in downstream analyses.”

R3.10.

It needs to be clear that figure 1 is not an exhaustive representation of all the ways that participation bias can arise in a genetic study.

We have now extended the legend of figure 1 to make clearer that the illustration does not present all possible scenarios of how selective participation can lead to bias in genetic studies.

Legend (Figure 1): "The figure is a simplified illustration of how participation bias can impact results obtained from two commonly employed methods in genomic studies. For further examples illustrating the impact of selection bias see Hernán et al. (2004)⁷."

7. Hernán MA, Hernández-Díaz S, Robins JM. A structural approach to selection bias. *Epidemiology*. 2004;15(5):615-625. doi:10.1097/01.ede.0000135174.63482.43

Decision Letter, first revision:

10th January 2023

Dear Dr. Schoeler,

Thank you for submitting your revised manuscript "The impact of participation bias in the UK Biobank on genetic associations and downstream analyses" (NATHUMBEHAV-22092529A). It has now been seen by the original referees and their comments are below. As you can see, Reviewers 1 and 2 are now satisfied with your revisions and recommend acceptance of your paper. However, Reviewer 3 still has fundamental concerns about your claims and conclusions, which do not match the evidence you present. These are important concerns that must be addressed thoroughly and with extensive changes to your manuscript throughout the text. If Reviewer 3's comments are addressed in full and with due care; and Reviewer 1's request to make the weights publicly available is met, we will be happy to accept your paper for publication.

In addition to the reviewers' comments, we also have a number of requests as part of finalizing your work for publication. We are now performing detailed checks on your paper and will send you a checklist detailing our editorial and formatting requirements within a week. Please do not upload the final materials and make any revisions until you receive this additional information from us.

Sincerely,

Arunas Radzvilavicius, PhD
Editor, Nature Human Behaviour

Nature Research

Reviewer #1 (Remarks to the Author):

The authors addressed all the comments. I strongly recommend the weights generated as part of this work be made available to other researchers via the UK Biobank portal.

Reviewer #2 (Remarks to the Author):

I thank the authors for thoroughly responding to and addressing my comments and concerns from round 1 or reviews. I have no further comments or recommendations at this time.

Reviewer #3 (Remarks to the Author):

1. My previous concern that the results of the paper are being over-sold has not been fully allayed. In practice, for any given analysis, you have only two results: the weighted and the unweighted. Comparing the two, you cannot tell (without external information) which is less biased, or in what direction the bias goes. I have played around with simulations and can simulate scenarios where the weighted analysis is more biased than the unweighted, and where different weighting models lead to estimates that are either biased towards or away from the null (for the same dataset). So I don't think that the results provide a "lower bound" for bias, and should not be described in this way. Nor do I think that you can say that the weighting "reduces" participation bias. You cannot know this, in general.

2. This problem implies more amendment than a paragraph in the discussion. It is crucial to the whole paper.

3. As with my comment at the first review, I would prefer to see the paper presented more as a sensitivity analysis for selection bias, rather than an absolute removal of the bias. I.e. the "truth" of the weighted results still depends on the assumption that the weighting model is correct, and this cannot be verified given the observed data. But this shouldn't be relegated to a paragraph in the discussion.

4. For example, in the abstract, I don't think you can say "revealed biases in SNP estimation in both directions" (because you really can't know what direction the true bias is in), or "identifying novel SNP associations for 12 traits" as these could be biased. I think fine to say e.g. "heritability estimates less impacted by weighting". The rest is fine, as not claiming any particular truth – I think that your point is that participation can distort results, and weighting can change results (showing they are sensitive to participation) – and you might think that on the whole the weighted results are likely to be less biased than the unweighted ones, but without external information you really cannot tell for sure.

5. On page 6 though you can talk about bias reduction, as these are the associations where you do know the "truth". Although this only applies if you assume that the association in the reference sample is "true" – for example if there is unmeasured confounding of X and Y by U, and selection into UKBB depends on U, then actually the estimate from UKBB will be less biased than the association in the reference sample. So I think it is important to point out that your definition of bias is not wrt the actual truth, but only wrt the reference population. You do give the definition, "Here, we quantified participation bias as the difference between an estimate of association

obtained in the UKBB (rUKBB) and the reference sample" but I think need to add to that sentence to emphasise that this is not the usual definition of bias, because you are not comparing to a known truth.

6. Page 6 "useful tool for bias reduction" – I would say useful tool for examining potential for bias.

7. The headings for the figures I think need amending too – with no truth to compare to, I don't think you can be sure there is under- or over-estimation due to participation bias.

8. The introduction could benefit from spelling out when the unweighted analysis is likely to be biased – i.e. an estimate of the regression coefficient of Y on X likely to be biased if selection into the sample is related to Y, conditional on X. This is implied by the figures but could be usefully stated.

9. I think that if an estimate changes between weighting and unweighting, then at least one of those estimates must be biased. If the estimate doesn't change, then that doesn't necessarily mean that both are unbiased - could also be that the weighting model is not correct, i.e. there are other factors related to the analysis model and selection that have not been included in the weighting model. if you agree with this intuition then might be worth saying something like this in the paper, so that users know how to interpret the results they get when using this method in practice.

Final Decision Letter:

Dear Dr Schoeler,

We are pleased to inform you that your Article "Participation bias in the UK Biobank distorts genetic associations and downstream analyses", has now been accepted for publication in *Nature Human Behaviour*.

Please note that *Nature Human Behaviour* is a Transformative Journal (TJ). Authors whose manuscript was submitted on or after January 1st, 2021, may publish their research with us through the traditional subscription access route or make their paper immediately open access through payment of an article-processing charge (APC). Authors will not be required to make a final decision about access to their article until it has been accepted. IMPORTANT NOTE: Articles submitted before January 1st, 2021, are not eligible for Open Access publication. Find out more about Transformative Journals

Once your manuscript is typeset and you have completed the appropriate grant of rights, you will receive a link to your electronic proof via email with a request to make any corrections within 48 hours. If, when you receive your proof, you cannot meet this deadline, please inform us at risproduction@springernature.com immediately. Once your paper has been scheduled for online publication, the Nature press office will be in touch to confirm the details.

With best regards,

Arunas Radzvilavicius, PhD
Editor, Nature Human Behaviour
Nature Research